# JAK-STAT-dependent contact between follicle cells and the oocyte controls *Drosophila* anterior-posterior polarity and germline development

Charlotte Mallart [1], Sophie Netter[2,4], Fabienne Chalvet[1,4], Sandra Claret [3], Antoine Guichet [3], Jacques Montagne[1], Anne-Marie Pret[2] & Marianne Malartre [1] ✉

The number of embryonic primordial germ cells in *Drosophila* is determined by the quantity of germ plasm, whose assembly starts in the posterior region of the oocyte during oogenesis. Here, we report that extending JAK-STAT activity in the posterior somatic follicular epithelium leads to an excess of primordial germ cells in the future embryo. We show that JAK-STAT signaling is necessary for the differentiation of approximately 20 specialized follicle cells maintaining tight contact with the oocyte. These cells define, in the underlying posterior oocyte cortex, the anchoring of the germ cell determinant *oskar* mRNA. We reveal that the apical surface of these posterior anchoring cells extends long filopodia penetrating the oocyte. We identify two JAK-STAT targets in these cells that are each sufficient to extend the zone of contact with the oocyte, thereby leading to production of extra primordial germ cells. JAK-STAT signaling thus determines a fixed number of posterior anchoring cells required for anterior-posterior oocyte polarity and for the development of the future germline.

Primordial germ cells (PGCs) are the precursors to the germline, which give rise to eggs and sperm in all sexually reproducing animals. Characteristic to all germ cells are germ granules, composing the germ plasm[1]. In *Drosophila*, germ plasm, assembled during oogenesis at the posterior pole of the oocyte, is essential for the formation of embryonic PGCs and for body axis polarity[2]. The number of PGCs is determined by the amount of germ plasm formed in the posterior region of the oocyte. Indeed, defects in germ plasm assembly result in decreased PGC number[3], while an excess of PGCs has been reported in embryos presenting extra copies of the maternal determinant *oskar* gene, which is essential for germ plasm assembly[4,5]. During oogenesis, *oskar* mRNA transport towards the posterior region of the oocyte

starts at stage 8, which is also an essential step in establishing anterior-posterior polarity[6]. When reaching the posterior cortex, *oskar* mRNA is translated into an RNA-binding protein, thereby recruiting other components, such as *nanos* (*nos*) mRNA, all together forming the germ plasm[7]. Consequently, both *oskar* and *nos* mutant embryos lack germ plasm, fail to produce PGCs and do not develop abdominal segments[2,6]. Defining an appropriate number of PGCs is important in all species, but how this number is determined is not fully understood.

*Drosophila* ovaries are composed of several ovarioles, which are strings of follicles (or egg chambers), connected by interfollicular stalks, progressively maturing through 14 stages, ending with the egg. Each follicle consists of a germline cyst made up of 16 germ cells (GCs),

[1]Institute for Integrative Biology of the Cell (I2BC), CEA, CNRS, Université Paris-Saclay, Gif-sur-Yvette, France. [2]Institute for Integrative Biology of the Cell (I2BC), CEA, CNRS, Université de Versailles-Saint-Quentin en Yvelines, Université Paris-Saclay, Gif- sur-Yvette, France. [3]Université Paris Cité, CNRS, Institut Jacques Monod, Paris, France. [4]These authors contributed equally: Sophie Netter, Fabienne Chalvet. ✉e-mail: marianne.malartre@universite-paris-saclay.fr

the oocyte and 15 nurse cells, surrounded by follicle cells (FCs) forming a monolayered epithelium with a group of polar cells (PCs) at the anterior and posterior poles. In the anterior region of follicles, FCs differentiate into several subpopulations at stage 8 of oogenesis in response to a gradient of JAK-STAT activity created by diffusion of the Unpaired (Upd) ligand, which is secreted from PCs[8]. In the posterior region, the EGFR ligand Gurken (Grk) secreted by the oocyte is received by adjacent FCs, which, in combination with JAK-STAT signaling, triggers posterior follicle cell (PFC) differentiation at stage 6[9–12]. Communication between FCs and GCs is necessary for correct follicle development[13]. For instance, it is thought that soon after their differentiation, PFCs send a signal back to the oocyte, which is necessary for the polarized transport of *oskar* mRNA[10,11,14,15]. The nature of this signal has not been identified to date but it has been proposed that it must be sharply localized within the follicular epithelium[12,16–18]. At stage 7, before *oskar* mRNA transport is initiated, non-muscle Myosin II is activated at the posterior oocyte cortex[19]. This activation is achieved by the di-phosphorylation of the myosin regulatory light chain (MRLC), and is required for the recruitment and maintenance of the evolutionarily-conserved Par-1 polarity protein. Par-1 then inhibits the presence of Shot/Patronin-dependent microtubule minus-end foci at the oocyte posterior cortex, thereby triggering a slight bias in microtubule plus-end focusing towards the posterior pole of the oocyte, which is sufficient for polarized *oskar* mRNA transport[20–23]. This reorganization of the oocyte microtubule network is therefore an early event in the establishment of anterior-posterior polarity[24–26]. Kinesin heavy chain (Khc), the force generating subunit of the microtubule motor protein Kinesin-1, participates to the transport of *oskar* mRNA and can thus also be detected at the oocyte posterior cortex[27]. Finally, the actin cytoskeleton plays a role in *oskar* mRNA anchoring. In particular, the actin motor protein Myosin-V links the *oskar* mRNA transport complex to the oocyte posterior cortex, participating to its entrapment[28,29]. Recently, the importance of physical contact between the oocyte and FCs for posterior Par1 and *oskar* mRNA localization has been reported at stage 9, suggesting that the follicular epithelium could be important for oocyte anterior-posterior polarity during several stages[30].

If the mechanisms occurring within the oocyte leading to *oskar* mRNA posterior transport and anchoring have been described extensively (reviewed in[31,32]), how the adequate amount of *oskar* mRNA is localized at the posterior oocyte cortex to later allow an appropriate number of PGCs to form in the embryo remains unknown. Here, we show that a population of 20 cells on average in the posterior-most follicular epithelium, which we named the posterior anchoring cells (PACs), defines the size of the *oskar* mRNA anchoring zone in the adjacent oocyte. We found that JAK-STAT activity determines the number of PACs, subsequently limiting the number of PGCs in the embryo. Importantly, we demonstrate that the PACs are the only FCs to maintain a tight contact with the oocyte, which is required to focus *oskar* mRNA polarized transport. Our results suggest that, in PACs, the junctional protein E-Cadherin and filopodia participate in keeping the contact between the two tissues. Our findings highlight the crucial role of the follicular epithelium during oogenesis for the formation of an appropriate number of PGCs in the future embryo, thus ensuring reproductive success.

## Results

### JAK-STAT signaling in follicles controls the future germline
In a previous study, we explored the regulation of JAK-STAT signaling in the follicular epithelium during oogenesis[33]. This led us to consider here whether misregulation of JAK-STAT signaling during oogenesis could affect early embryonic development. By expressing the dominant gain-of-function *hopscotch*[tumoral] (*hop*[TUM]) allele[34,35] in PFCs with the *E4-Gal4* driver (Supplementary Fig. 1), we found that fertilized eggs arising from these follicles contain an excess of PGCs, identified with

Vasa immunostaining, in stages 5 to 12 embryos (Fig. 1A–B). Hence, JAK-STAT misregulation in somatic cells during oogenesis affects germline development of the progeny. Up to now, only gain of *oskar* copies has been reported to cause an excess of PGCs in the early embryo[4,5]. We thus examined the localization of *oskar* mRNA at stage 10a, when it can be easily detected (Fig. 1C,D), in follicles presenting JAK-STAT overactivity as compared to controls. *oskar* mRNA is transported with Staufen at the posterior oocyte cortex where it is translated. Hence, antibody staining against Oskar or Staufen can be used as a proxy for *oskar* mRNA localization[36–38]. Since follicle size varies greatly at this stage with a constant number of cells, we counted the number of FCs facing the accumulation of Staufen in the oocyte as a means to quantify the size of the *oskar* mRNA anchoring zone. In controls, we found that *oskar* mRNA is localized in an oocyte region of robust size, which restricts between stages 9 and 10, to face consistently two FCs around PCs in all directions at stage 10 (Fig. 1E). In follicles upregulating JAK-STAT signaling however, the size of the *oskar* mRNA anchoring zone was significantly increased, which we also confirmed by quantifying the diameter of Staufen accumulation (Fig. 1F, G). Similar results were obtained with Oskar protein, confirming that sustained JAK-STAT activity in the posterior follicular epithelium is sufficient to increase the size of the *oskar* mRNA anchoring zone in the adjacent oocyte (Supplementary Fig. 2A, B).

JAK-STAT signaling forms a gradient with the highest activity at follicle poles where Upd-secreting PCs are located[12]. By using a JAK-STAT activity reporter[39], we previously showed that this gradient becomes asymmetric between anterior and posterior poles, as JAK-STAT activity progressively restricts between stage 7 and late stage 8 to a smaller number of PFCs[33]. Since our results indicate that increased JAK-STAT signaling during oogenesis affects the development of the future germline, we sought to identify precisely which FCs display JAK-STAT activity in control follicles at stage 10. Between stages 8 and 9, JAK-STAT signaling at posterior poles restricts to a region of robust size of only two FCs around PCs in all directions, which is maintained at stage 10 (Fig. 1H, I). These cells are a subpopulation of PFCs, since *pointed-LacZ (pnt-LacZ)*, a PFC marker, can be detected in a much larger region forming, on average, a diameter of 10 cells with PCs at the center at stage 10 (Supplementary Fig. 3A–C'). When observed in a 3D reconstruction of the follicle from a pole view, the PFC subpopulation displaying high JAK-STAT activity forms two concentric circles of FCs around PCs corresponding to a total population of 19.3 cells on average (Fig. 1H', H'', J). To verify that targeting *hop*[TUM] to PFCs indeed translates into an increase in JAK-STAT activity, we quantified the number of GFP positive cells in flies expressing both the *10XSTAT92E-GFP* activity reporter and *hop*[TUM]. We found that, in sectional views, an average of 10 cells around PCs display high JAK-STAT activity instead of only 4 in control follicles, indicating that upon *hop*[TUM] expression, all PFCs become highly active for JAK-STAT signaling (Fig. 1K', L). Increasing the size of the domain of JAK-STAT activity therefore correlates with an increase in the *oskar* mRNA anchoring zone. Note that while the correlation is strict in control follicles, the size of JAK-STAT activity in FCs exceeds the size of *oskar* mRNA anchoring in the gain of JAK-STAT activity condition. (Fig. 1K-K'). Altogether, our results demonstrate that JAK-STAT activity in the 20 posterior-most PFCs delimits the zone of *oskar* mRNA anchoring in the oocyte, which is crucial for restricting the number of PGCs in the future embryo.

### EGFR signaling acts on E-Cadherin to restrict JAK-STAT activity
We next wondered how the boundary of JAK-STAT activity is sharply delimited to a fixed number of PFCs at stage 9 from the smooth gradient observed in earlier stages. We previously showed that E-Cadherin regulates JAK-STAT signaling positively and that both JAK-STAT activity and *shg* expression follow a similar dynamics during oogenesis, leading to their progressive restriction at posterior poles of follicles[33]. Indeed, *shg* expression, as visualized by a *shg-LacZ*

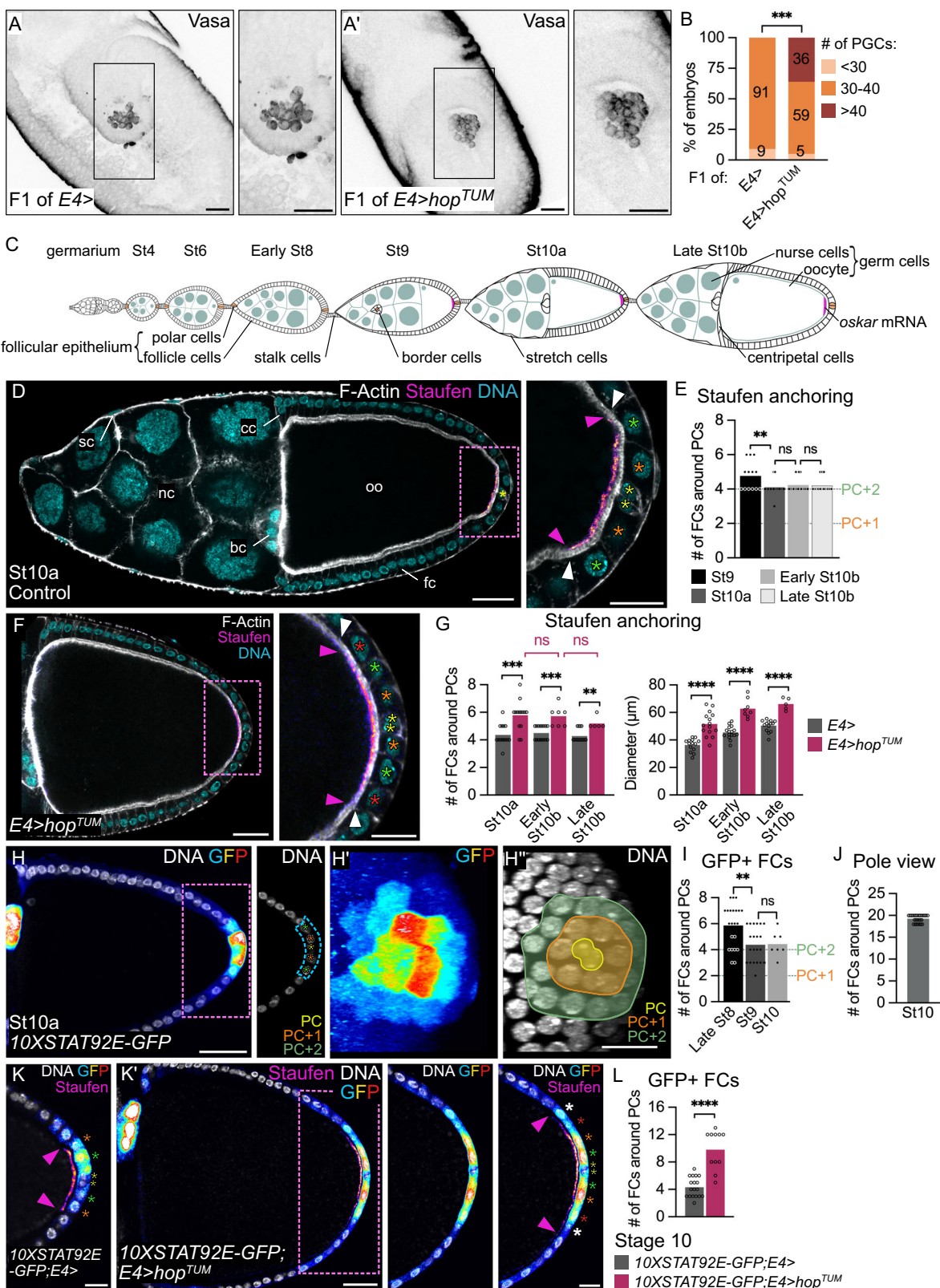

transcriptional reporter and E-Cadherin accumulation, decreases in the posterior follicular epithelium surrounding the oocyte from stage 7, and this decrease becomes particularly striking at stage 8 (Mallart et al.[33], Supplementary Fig. 3D, E, Fig. 2A). Hence, all PFCs, defined by the expression the *pnt* marker (Supplementary Fig. 3A), display low E-Cadherin levels. Since PFCs overlie the oocyte, it is possible that the oocyte could be the source of a signal inhibiting *shg* expression. To test this hypothesis, we analyzed follicles expressing an RNAi targeting *shg* mRNA in GCs using the *nos-Gal4* driver. Resulting follicles contain oocytes that are able to signal to neighboring FCs but are mispositioned, since *shg* is required to anchor the oocyte at the posterior pole of follicles[40,41]. Remarkably, we observed a decrease in E-Cadherin levels in FCs surrounding the oocyte, wherever it was mislocalized to in the egg chamber, in 100% of cases, indicating that oocyte proximity

**Fig. 1 | Restricted JAK-STAT signaling activity to 20 PFCs during oogenesis defines PGC number in the future embryo.** Posterior is to the right. **A,A'** Stage 9 embryos arising from control follicles and follicles expressing *hop^TUM* in PFCs under the control of *E4-Gal4*. Max z-projection showing Vasa (black) to identify PGCs. **B** Quantification of PGC number in stage 5 to 12 embryos. Numbers within bars indicate the percentage of embryos in each category. **C** Schematic drawing of an ovariole. The different germ and somatic cell types are indicated. *oskar* mRNA localization appears in pink. **D, F, H, K** Posterior-most follicle regions are magnified to focus on *oskar* mRNA anchoring zone (pink dotted rectangles). Yellow, orange, green, red and white stars indicate PCs, PC + 1, +2, +3 and +4 rows respectively. Arrowheads indicate the anchoring zone boundary on oocyte (pink) and FC (white) sides. **D** Control follicle (*fruitless-Gal4*/+) stained for F-Actin (gray, phalloidin), DNA (cyan, DAPI) and Staufen (fire filter). **E** Measure of the number of FCs around PCs facing Staufen signal. **F** Follicle expressing *hop^TUM* under the control of *E4-Gal4*. **G** Anchoring zone size quantified as the number of FCs around PCs facing Staufen

signal, and in diameter. **H, K, K'** Sum z-projection of 5 consecutive 1 µm confocal sections of follicles expressing *10XSTAT92E-GFP*, stained for DNA (gray, DAPI), GFP (royal filter) and, for (**K,K'**), Staufen (fire filter). JAK-STAT activity is presented in section view of the posterior pole, outlined in blue (**H**), and in 3D projections of GFP and DNA staining (**H',H''**), showing the first FC row around PCs in orange (PC + 1) and the second in green (PC + 2). Quantifications of the number of GFP-positive (GFP + ) PFCs on section views (**I, L**) and of FCs around PCs on 3D projections of DNA staining (pole view, **J**). #: number. St stage, nc nurse cells, oo oocyte, fc follicle cells, bc border cells, sc stretch cells, cc centripetal cells. Scale: 30 µm except magnifications of (**D**–**K'**) 15 µm. Statistical tests: two-sided Fisher's exact test for (**B**); two-sided Mann–Whitney for (**E, G**) # of FCs around PCs; two-sided unpaired *t* test for (**G**) Diameter and (**I, L**). Information about statistics and reproducibility is provided in Supplementary Tables 1–6. Source data are provided as a Source Data file.

inhibits *shg* expression in the overlying follicular epithelium (Fig. 2B). Grk is a known ligand diffusing from the oocyte to activate EGFR signaling in FCs from stage 6, thereby triggering PFC differentiation[10,11]. We tested whether Grk could also be responsible for the decrease in *shg* expression in PFCs. In 100% of *grk* mutant follicles, we observed that the E-Cadherin posterior decrease did not occur, indicating that EGFR signaling inhibits E-Cadherin accumulation in PFCs (Fig. 2C). To confirm this result, we decreased EGFR activity in mosaic clones of PFCs expressing RNAi targeting *EGFR* mRNA and observed that E-Cadherin levels were high in all clones compared to control neighboring PFCs in a cell-autonomous manner (Fig. 2D). EGFR signaling is therefore a negative regulator of E-Cadherin levels in PFCs from stage 7 onward, and this regulation is likely to occur at the transcriptional level, since *shg* transcription is downregulated in PFCs in control follicles (Supplementary Fig. 3D–G).

Interestingly, in this low E-Cadherin environment, we found that *shg* reporter expression (Supplementary Fig. 3D–H) and E-Cadherin levels actually increase progressively from stage 7 in a few cells only, to reach two FCs around PCs in all directions by stage 10 (Fig. 2E, F). This corresponds exactly to the same population of FCs displaying high JAK-STAT activity. We thus assessed whether *shg* could be a JAK-STAT signaling target in these cells by using RNAi-mediated *upd* knockdown in PCs to decrease JAK-STAT activity in the follicular epithelium. We observed a decrease in the expression of the *shg-LacZ* transcriptional reporter in the posterior-most PFCs, indicating that JAK-STAT signaling is required to activate *shg* expression in these cells (Fig. 2E,E' and Supplementary Fig. 3H). We performed the reverse experiment by directing the expression of constitutively active *hop^TUM* in all PFCs with the *E4-Gal4* driver and found more PFCs with high E-Cadherin levels than in controls, showing that JAK-STAT signaling regulates E-Cadherin levels positively (Fig. 2F,F'). Together, our results demonstrate that JAK-STAT signaling is necessary and sufficient for *shg* expression in the posterior-most PFCs between stages 7 and 10. Hence, a positive feedback loop is created by E-Cadherin being both an activator[33] and target of the JAK-STAT pathway. This could contribute to establish the sharp border of JAK-STAT activity in the posterior-most PFCs. Altogether, our findings thus reveal that JAK-STAT and EGFR signaling pathways have opposite effects on *shg* expression (Fig. 2G). This interplay restricts JAK-STAT activity to a subpopulation of PFCs and creates a sharp boundary with surrounding PFCs devoid of JAK-STAT activity.

### A subset of JAK-STAT active PFCs contacts the oocyte

JAK-STAT activity is thus restrained to a fixed number of PFCs and *oskar* mRNA is precisely localized to a region of the oocyte cortex facing exactly these PFCs. Interestingly, the border of the zone of *oskar* mRNA anchoring stops exactly in front of FC-FC junctions, indicating that *oskar* mRNA is anchored in a domain facing the entire apical region of a defined number of FCs (Fig. 3A, white arrowheads). Strikingly, we observed that these PFCs maintain a particular contact with the

oocyte, until later than their more lateral PFC neighbors (Fig. 3A). Indeed, until early stage 9, the oocyte and all adjacent FCs are in close contact (Supplementary Fig. 4A). Then the extracellular space between membranes increases progressively as vitelline components secreted apically by FCs accumulate[42–44]. We measured the size of the perivitelline space between stages 9 and 10 and found that oocyte and FC membranes do not separate uniformly (Fig. 3B, B' and Supplementary Fig. 4A–D). The perivitelline space becomes visible by confocal microscopy at late stage 9 along the lateral sides of the oocyte-FC interface, while posterior oocyte and FC membranes remain tightly associated (Supplementary Fig. 4B), the difference being accentuated at stage 10a (Fig. 3A). The perivitelline space finally becomes homogenous all around the oocyte by late stage 10b (Supplementary Fig. 4D). We have therefore identified a subpopulation of PFCs with high JAK-STAT activity consistently facing the *oskar* mRNA anchoring zone and presenting a tight contact with the oocyte at stages 9 and 10 of oogenesis.

### JAK-STAT signaling is required for *oskar* mRNA localization

It has been shown previously that Staufen posterior localization in the oocyte is lost in front of PFCs defective for JAK-STAT activity[12]. In order to examine the role of JAK-STAT signaling more specifically on the PFC-oocyte interface, we knocked down *upd* in PCs. We found that Staufen posterior localization was affected in 63% of stage 10 follicles, with phenotype strengths ranging from a reduced Staufen crescent to partially anchored or even fully unanchored Staufen (Fig. 3A,A' and Supplementary Fig. 5A, B). Similar results were obtained with Oskar::GFP as a read-out, which is detected in front of the same number of PFCs as the endogenous Oskar protein at stage 10 (Supplementary Fig. 2A, B and Supplementary Fig. 5C, E). In addition, we found that Oskar::GFP mislocalization phenotypes were maintained until the end of oogenesis (Supplementary Fig. 5D,F,G). Remarkably, although the oocyte membrane remains in tight contact with the posterior-most PFCs until stage 10b in controls, we observed that they were already separated at stage 9 when JAK-STAT signaling was decreased and the perivitelline space was significantly wider at stages 9 and 10a (Fig. 3A,A',C,C'). These results therefore indicate that JAK-STAT signaling is required to keep a subpopulation of PFCs tightly associated to the oocyte. Accordingly, contact between FCs and the oocyte was recently shown to be a requirement for *oskar* mRNA posterior localization using physical separation methods[30]. Here, we have identified the PFC subpopulation that maintains a tight contact with the oocyte, allowing correct *oskar* mRNA anchoring to the posterior oocyte cortex. We named these cells the posterior anchoring cells (PACs). We next measured the space between PAC and oocyte membranes in follicles displaying increased JAK-STAT activity by expressing the *hop^TUM* allele in PFCs. Importantly, as opposed to what occurs when JAK-STAT activity is reduced, the tight contact between PAC and the oocyte membranes was maintained in a larger region upon JAK-STAT

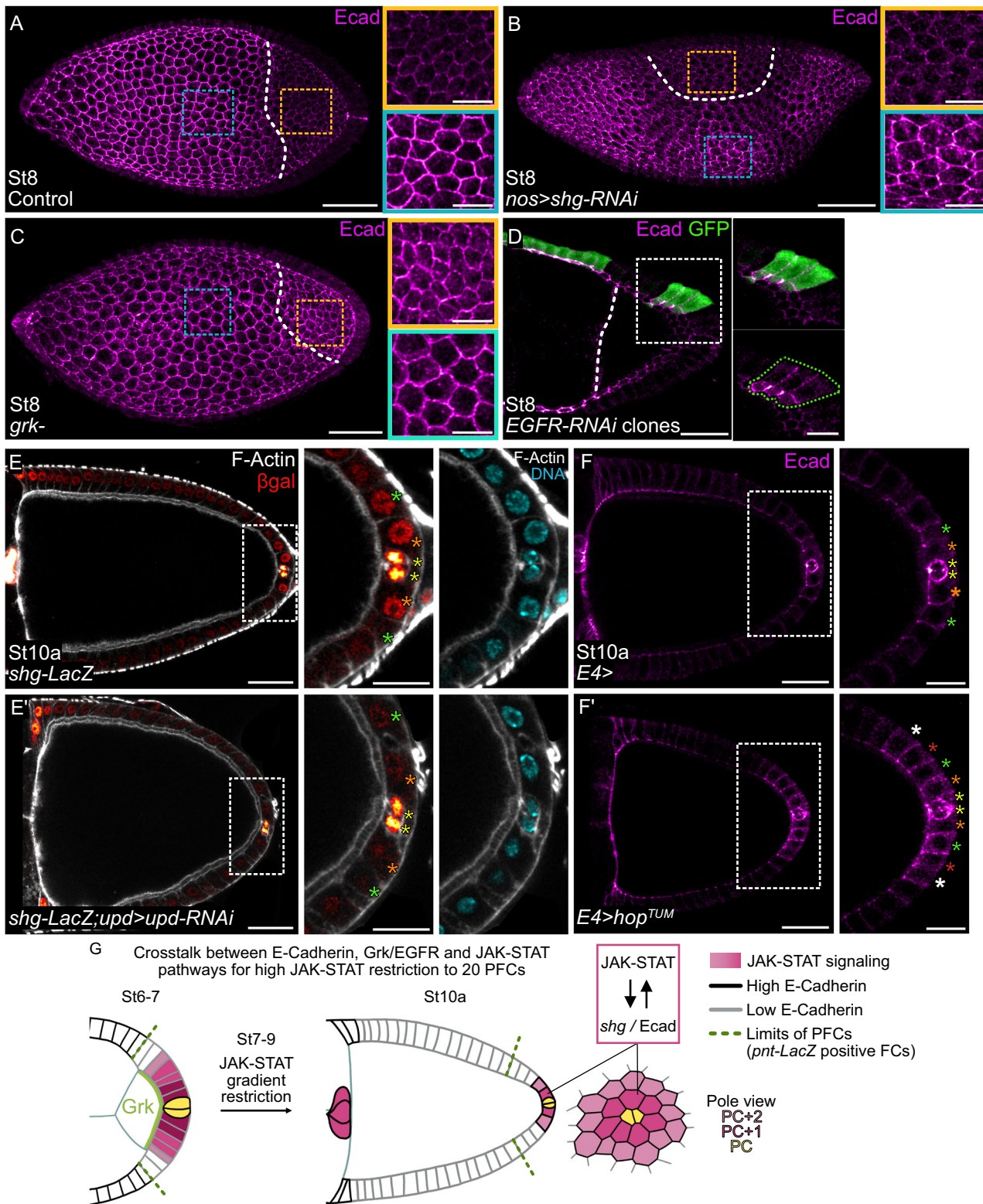

signaling gain-of-function as compared to that of controls (Fig. 3A", D). Interestingly, we observed a direct correlation between JAK-STAT activity, as assessed with the 10XSTAT92E::GFP reporter, and the follicle cell/oocyte membrane space. Taking advantage of the mosaicism of the *E4-Gal4* driver, which is active in PFCs and some lateral FCs (Supplementary Fig. 1), we found tight contact between follicle and oocyte membranes only when FCs exhibited constitutive JAK-STAT activity (Fig. 3E). Strikingly, the tight contact was observed even outside the PFC domain, i.e. beyond PC + 5 (Supplementary Fig. 3C').

Indeed, *hop^{TUM}* is sufficient to maintain FC and oocyte membranes tighter than their non-JAK-STAT active neighbors as far as PC + 10 (Fig. 3E, inset). Note that this is not however sufficient to localize *oskar* mRNA that far from the posterior pole. Together, our results demonstrate that JAK-STAT activity is both necessary and sufficient to maintain PAC and oocyte membranes tight. Restricted JAK-STAT activity is thus required in the posterior follicular epithelium to define precisely the size of *oskar* mRNA anchoring zone by keeping a specific contact with the adjacent oocyte (Fig. 3F).

**Fig. 2 | Opposite actions of EGFR and JAK-STAT signaling pathways restrict E-Cadherin levels to a subpopulation of PFCs.** Posterior is to the right. **A**–**D** Max z-projections of E-Cadherin (Ecad, magenta hot filter) immunostaining on stage 8 follicles. The white dotted lines indicate the nurse-oocyte frontier. **A**–**C** Yellow and blue dotted rectangles indicate magnified FCs in contact with the oocyte and further from it, respectively. **A** Control follicle (*shg-LacZ*). **B** Follicle expressing *shg-RNAi* in germ cells under the control of *nos-Gal4* resulting in a mispositioned oocyte. **C** *grk* mutant follicle showing high levels of E-Cadherin in FCs around the oocyte. **D** Mosaic clones (GFP positive, green) expressing *EGFR-RNAi* (#JF01084 line, similar phenotypes were obtained with #KK100051 line). White dotted square indicates the magnified zone, green dotted line outlines a PFC clone presenting high E-Cadherin levels. **B**–**D** 100% of stage 7-10 follicles analyzed presented similar phenotypes. **E-F'** Posterior-most PFCs are magnified (white dotted rectangles).

Yellow, orange, green, red and white stars indicate PCs, PC + 1, +2, +3 and +4 rows respectively. **E,E'** *shg-LacZ* reporter (hemizygous) in control follicle and in follicle expressing *upd-RNAi* in PCs under the control of *upd-Gal4* stained for F-Actin (gray, phalloidin), DNA (cyan, DAPI) and beta-galactosidase (βgal, red hot filter). See Supplementary Fig. 3 for quantification. **F,F'** Max z-projections of 4 consecutive 1 μm confocal sections. Control follicle and follicle expressing *hop^TUM* in PFCs under the control of *E4-Gal4* stained for E-Cadherin (magenta hot filter). **G** Schematic representation of the crosstalk between Grk/EGFR and E-Cadherin to restrict JAK-STAT activity to a subpopulation of 20 PFCs highlighted in the pole view. Subsequently, *shg* expression is induced in those cells, creating a positive feedback loop. St: stage. Scale: 30 μm except magnifications 10 μm for (**A**–**D**) and 15 μm for (**E-F'**). Information about reproducibility is provided in Supplementary Table 7.

## JAK-STAT is required in PFCs for microtubule polarity in the oocyte

To find out whether JAK-STAT signaling activity in PACs only regulates the anchoring of *oskar* mRNA at the posterior oocyte cortex or if it is required earlier, during the *oskar* mRNA transport phase, we examined the status of Myosin II in the oocyte. Indeed, Myosin II activation is the earliest identified marker of anterior-posterior polarity establishment occurring in the oocyte in response to back signaling from PFCs upon their differentiation[19]. It is necessary for Par1 posterior localization, which triggers microtubule network reorganization[21,22]. The activation status of Myosin II in the oocyte can be assessed with an antibody that recognizes MRLC in its di-phosphorylated (MRLC-2P) form only[45]. Interestingly, the di-phosphorylation of MRLC is dynamic during oogenesis[19] and is restricted between stages 7 and 10 to the posterior-most region of the oocyte cortex facing the PACs (Supplementary Fig. 6A, B). We found that MRLC-2P distribution at the oocyte posterior cortex at stage 10 was almost completely absent in *upd* knockdown follicles, in which PAC and oocyte membranes are precociously separated (Fig. 4A,A',B). Conversely, MRLC-2P staining was present in front of a greater number of cells when constitutively-active *hop^TUM* was expressed ectopically in PFCs (Fig. 4A,A",C). Remarkably, MRLC-2P accumulates specifically in the region where PAC and oocyte membranes are tightly associated and is absent as of the boundary where separation of these membranes becomes visible, both in control and JAK-STAT gain-of-function follicles (Fig. 4A,A", white arrowheads). Together, these results indicate that JAK-STAT signaling is necessary and sufficient in the follicular epithelium for MRLC di-phosphorylation at the posterior oocyte cortex, which is spatially correlated with PAC-oocyte membrane contact. Our results thus suggest that JAK-STAT signaling acts upstream of microtubule organization.

We next used the Khc::β-Galactosidase (Khc::β-Gal) fusion protein[46] to confirm that JAK-STAT signaling regulates oocyte microtubule polarity and thereby *oskar* mRNA transport, since Khc is required for transporting *oskar* mRNA towards microtubule plus-ends at the posterior pole of the oocyte[27,47]. According to its function, Khc::β-Gal localization in the oocyte follows the same dynamics as *oskar* mRNA with a progressive restriction between stages 9 and 10 until it is found only in front of the PACs (Fig. 4D, E). Upon *upd* knockdown, we found a strong reduction, or even a complete absence of Khc::β-Gal cortical distribution, with only dispersed staining being detected in the oocyte cytoplasm (Fig. 4D', F). JAK-STAT signaling in PACs is thus required to regulate the polarity of the oocyte microtubule network necessary for *oskar* mRNA transport towards the posterior cortex.

## The JAK-STAT target *shg* is sufficient to produce ectopic PACs

Next, we sought to investigate the mechanism by which PACs keep tight contact with the oocyte longer than their PFC neighbors. Since *shg* is specifically expressed in PACs, we examined whether extending *shg* expression to all PFCs by using the *E4-Gal4* driver would have the same effect on the PAC-oocyte membrane interface as a gain of JAK-STAT activity. We found that *shg* ectopic expression in PFCs leads to an increase in the region of tight contact between the follicular epithelium and the oocyte (Fig. 5A,A',C), even when mosaic clones expressing *shg* are found beyond the PFC domain, as far as PC + 11 (Fig. 5B,B'). The sizes of the zones of *oskar* mRNA anchoring zone and of MRLC di-phosphorylation at the oocyte cortex were also increased upon *shg* ectopic expression in PFCs (Fig. 5A-A',D–F). These results indicate that *shg*, like JAK-STAT signaling, is sufficient to produce ectopic PACs.

We therefore wondered whether JAK-STAT signaling acts through *shg* to maintain PAC-oocyte membrane in tight contact. We combined the *E4-Gal4* and *upd-Gal4* drivers to trigger concurrent *shg* ectopic expression in PFCs and *upd* knockdown in PCs, respectively. PAC number and *oskar* mRNA localization defects that were observed upon *upd* knockdown only were largely rescued by *shg* expression (Supplementary Fig. 7A–C). Remarkably, the abnormally wide perivitelline space between PACs and the oocyte observed upon *upd* knockdown was also rescued by *shg* expression (Supplementary Fig. 7D). Our findings therefore demonstrate that *shg* acts downstream of the JAK-STAT signaling pathway to maintain PAC-oocyte membranes in tight contact, which is required for *oskar* mRNA posterior transport and anchoring.

Finally, since embryos arising from follicles with ectopic PACs due to extended JAK-STAT activity contain an excess of PGCs, we wondered whether *shg* ectopic expression in PFCs could also have similar consequences on the future embryo. We thus assessed the number of PGCs in fertilized embryos arising from follicles expressing *shg* in PFCs with the *E4-Gal4* driver by counting Vasa-positive cells. We found a significantly higher number of PGCs in embryos descending from ovarian follicles with *shg* ectopic expression than in control embryos, indicating that *shg* is sufficient not only to generate ectopic PACs but also to produce extra PGCs in embryos (Fig. 5G, H).

## PACs form filopodia-like structures extending towards the oocyte

While examining carefully the PAC-oocyte interface, we noticed that actin-rich projections visualized using phalloidin were present at the posterior oocyte cortex of stage 9 and 10 follicles and that they seemed longer in the region facing the PACs (Fig. 6A). F-actin projections have been described at the posterior cortex of oocytes at stage 10[48–50], and actin-rich microvilli are also present between the oocyte and FC membranes coming from both cell types[51,52]. To distinguish the origin of the actin projections present specifically at the PAC-oocyte interface, we expressed the membrane marker mCD8::GFP in FCs with the somatic *traffic jam-Gal4* driver and stained the actin cytoskeleton. Surprisingly, all actin projections observed at the oocyte posterior cortex in contact with PACs were GFP-positive, indicative of their somatic origin (Fig. 6A, green arrowheads). By contrast, actin protrusions from the oocyte lateral cortex were not GFP-positive, showing that they do not originate from the follicular epithelium (Fig. 6A, white arrowheads), and may correspond to a distinct actin network generated within the oocyte that is not found at the posterior

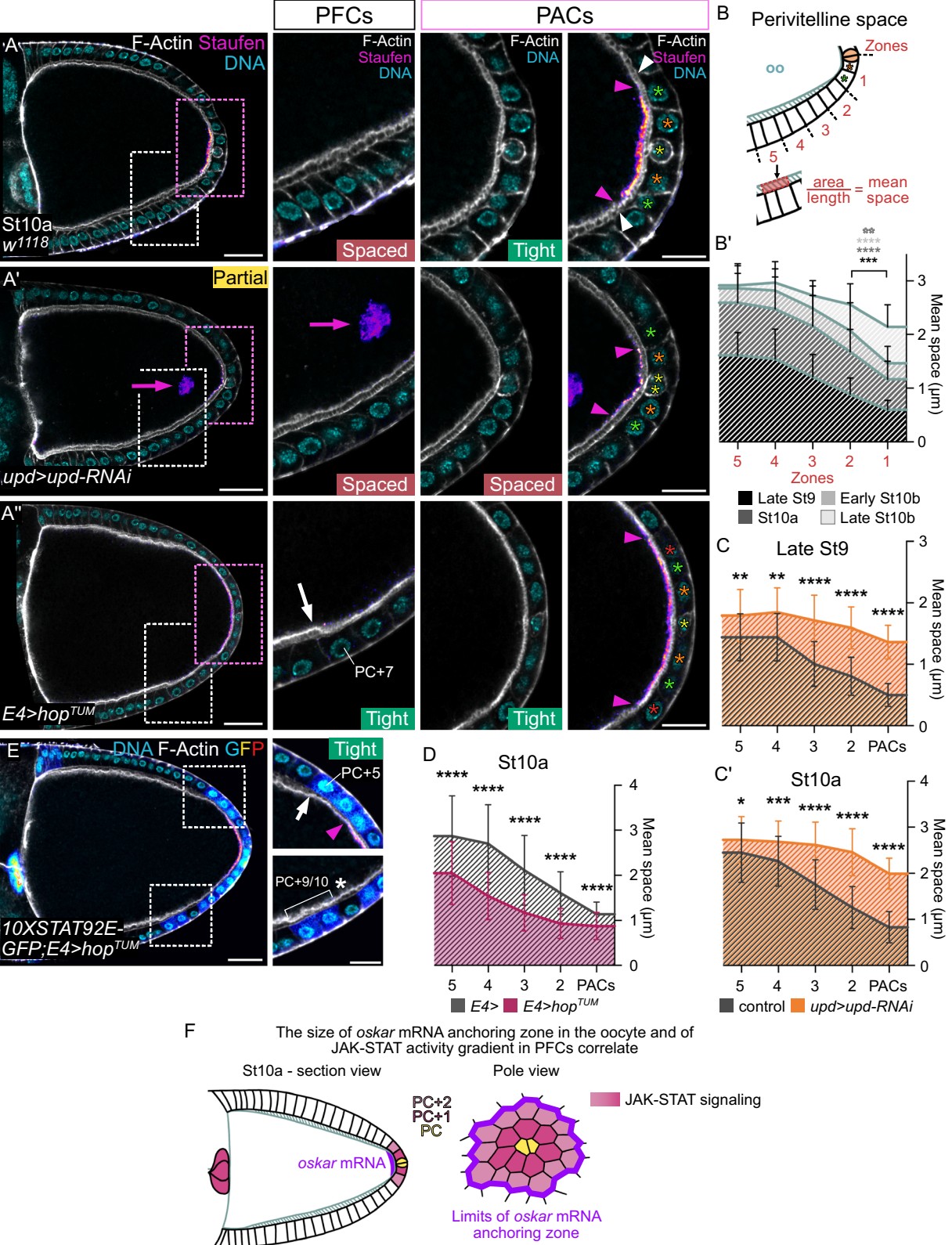

F The size of *oskar* mRNA anchoring zone in the oocyte and of
JAK-STAT activity gradient in PFCs correlate

cortex[53]. Our findings thus reveal that PACs form actin-rich protrusions penetrating the oocyte posterior cortex where *oskar* mRNA is located.

To characterize these projections emanating from the PACs at a higher resolution, we analyzed stage 9 follicles using electron microscopy. We used flies expressing mCD8::GFP in PFCs, together with a *UAS-GBP-APEX* transgene that allows visualization of GFP in electron microscopy. Since the activity of the ascorbate peroxidase (APEX)

forms electron-dense DAB precipitates when combined with an anti-GFP nanobody, any GFP-tagged protein can be detected by electron microscopy using this method[54]. We confirmed that long projections, up to 3 µm in length, extend from the PACs into the oocyte, crossing the perivitelline space where accumulating vitelline bodies can be visualized (Fig. 6B–C). We observed two apposed membranes outlining the projections, an inner PAC membrane appearing dark due to

**Fig. 3 | JAK-STAT activity in PACs maintains a tight contact with the oocyte and defines the size of *oskar* mRNA anchoring zone in the oocyte.** Posterior is to the right. Stage 10a follicles stained for F-Actin (gray, phalloidin), DNA (cyan, DAPI) and Staufen (fire filter) are shown. PACs (pink dotted rectangles) and adjacent PFCs (white dotted rectangles) are magnified. Yellow, orange, green and red stars indicate PCs, PC + 1, +2 and +3 rows respectively. Arrowheads indicate the anchoring zone boundary on oocyte (pink) and FC (white) sides. "Tight" and "Spaced" refer to the perivitelline space. In (**A″,E**), white arrows point to the limit of the increased region of tight oocyte-PFC contact. **A,A′,A″** Control follicle, and follicles expressing *upd-RNAi* in PCs under the control of *upd-Gal4* or *hop^TUM* in PFCs under the control of *E4-Gal4*. In (**A′**), "Partial" Staufen localization phenotype (smaller anchoring zone and detection in the oocyte cytoplasm, pink arrow) is shown. See Supplementary Fig. 5 for other phenotypes and quantifications. **B** Scheme of how the perivitelline space was measured and (**B′**) graph presenting these measurements made on control follicles (*fruitless-Gal4*/+). Data are presented as mean values ± SD. Gray stars indicate statistically significant differences between zones 1 and 2 (in red) of the corresponding stages. **C,C′,D** Measurements of the perivitelline space in control, *upd>upd-RNAi* and *E4>hop^TUM* follicles. Data are presented as mean values ± SD. **E** *E4>hop^TUM* follicle expressing *10XSTAT92E-GFP*. The white star highlights the wider perivitelline space in front of a JAK-STAT inactive FC in between active FCs. **F** Scheme representing the correlation between the *oskar* mRNA anchoring zone and JAK-STAT activity in section and pole views. Oo: oocyte. St: stage. Scale: 30 μm except magnifications 15 μm. Statistical tests: two-sided unpaired *t* test. Information about statistics and reproducibility is provided in Supplementary Tables 8–13. Source data are provided as a Source Data file.

the dense precipitates formed by the presence of mCD8::GFP, and a light outer oocyte membrane (Fig. 6B′-C). These extensions are close-ended and filled, which is consistent with the presence of F-actin structures (Fig. 6B′-C, magnifications). We wondered whether these posterior actin protrusions could participate in anchoring *oskar* mRNA particles by contacting them. We therefore used an Oskar::GFP construct that could be detected in electron microscopy using the GBP-APEX system, as a means to visualize *oskar* mRNA particles since they contain Oskar protein when localized at the posterior oocyte cortex[55]. This allowed us to detect Oskar particles in close proximity to filopodia originating from PACs, either apposed to filopodia on the same section or associated with filopodia on the next consecutive section (70 nm between sections) (Fig. 6D–E). Electron microscopy thus confirmed the existence of protrusions extending from the PACs into the posterior oocyte cortex, some of which coming into contact with particles containing Oskar, at stage 9 of oogenesis.

## JAK-STAT signaling is required for filopodia formation in PACs

We next investigated whether PACs express genes encoding proteins involved in filopodia formation among a list of genes expressed in the most posterior sub-population of follicle cells at late stage 8 identified by single-cell sequencing[56]. The Cdc42 downstream effector Enabled (Ena) was a promising candidate since it polymerizes actin monomers at the plus ends of actin filaments to form filopodia[57]. Remarkably, we found that Ena is produced at high levels specifically in PACs (Fig. 7A). Ena accumulates apically in PACs, at the interface with the oocyte, and can even be detected within filopodia (Fig. 7A, blue arrowheads). In addition, Ena accumulation follows the same restriction in PFCs between stages 7 and 10 as that of JAK-STAT activity (Supplementary Fig. 6C, D). To test whether Ena could be a target of JAK-STAT signaling, we knocked down *upd* and found that Ena did not accumulate in PACs (Fig. 7A,A′,B). Conversely, when JAK-STAT signaling was extended to all PFCs upon expression of *hop^TUM*, we found that the number of Ena-positive cells was significantly higher than in the control, and the region of the follicular epithelium elaborating filopodia was enlarged (Fig. 7A″,C, D). By generating clones of cells expressing *hop^TUM*, we found that JAK-STAT activity was sufficient to provide follicle cells with the ability of expressing Ena and elaborating filopodia even beyond the PFC domain (Fig. 7E). Indeed, control cells adjacent to these clones and closer to PCs did neither express Ena nor elaborate filopodia, indicating a cell autonomous effect of *hop^TUM* on these phenotypes. To further explore the link between JAK-STAT signaling and filopodia formation, we investigated whether filopodia formation depends on JAK-STAT activity. To test this hypothesis, we needed another filopodia marker than Ena since it is a target of JAK-STAT signaling. We therefore expressed *mCD8::GFP* in PFCs with the *E4-Gal4* driver, in follicles in which *upd* was knocked down by targeting *upd-RNAi* in PCs with the *upd-Gal4* driver. In that case, the proportion of actin protrusions in the oocyte that were also GFP positive, indicative of their somatic origin, was significantly decreased compared to control follicles, which display 100% of GFP positive actin protrusions at the PAC-oocyte

interface (Fig. 7F–G). This result demonstrates that JAK-STAT signaling is necessary for filopodia formation in PACs.

Altogether, our results identify Ena as a target of JAK-STAT signaling in PACs and show that JAK-STAT activity is both necessary and sufficient for the formation of filopodia in the posterior follicular epithelium (Fig. 7H).

Having identified two JAK-STAT signaling targets in PACs, E-Cadherin and Ena, we investigated a potential hierarchical relationship between these two proteins. We first expressed *ena* in all PFCs with the *E4-Gal4* driver, but we observed no change in E-Cadherin accumulation compared to control follicles in that condition, indicating that Ena cannot induce higher levels of E-Cadherin in these cells (Supplementary Fig. 8A–B). In contrast, we observed a higher number of cells positive for Ena upon *shg* ectopic expression (Supplementary Fig. 8C–D). This result could be interpreted in two ways. Either ectopic *shg* expression can upregulate Ena levels directly, or it increases JAK-STAT signaling in PFCs, which in turn induces *ena* expression. Indeed, we showed previously that E-Cadherin acts positively on JAK-STAT signaling up to stage 9[33]. We thus expressed *shg* in PFCs with the *E4-Gal4* driver and assessed JAK-STAT activity with the *10XSTAT92E-GFP* reporter in stage 10 follicles. We observed that under these conditions, JAK-STAT activity in PFCs is higher than in the control, indicating that *shg* regulates JAK-STAT signaling positively at the posterior poles of follicles at stage 10, as it does earlier in oogenesis (Supplementary Fig. 8E–F). Hence, although we cannot rule out a direct effect of *shg* on Ena levels, it is possible that the increase in Ena accumulation in PFCs observed upon *shg* expression in these cells is due to an indirect effect of *shg* on JAK-STAT signaling (Supplementary Fig. 8G).

## *enabled* is sufficient to keep PACs and the oocyte in contact

To test whether Ena could participate in *oskar* mRNA anchoring, we extended *ena* expression to all PFCs and found that the *oskar* mRNA anchoring zone at the oocyte posterior cortex was significantly enlarged at stage 10 (Fig. 8A–B). Strikingly, oocyte and PFC membranes also remained in tight contact in a larger region than in the control (Fig. 8A,A′,C). To assess whether there is a link between the presence of filopodia in FCs and the region of tight contact between FC and oocyte membranes, we examined the apical surface of PFCs upon ectopic *ena* expression. In this setting, we found that the region where FCs elaborate filopodia, which can be identified by the presence of both actin and Ena markers, was extended as compared to control follicles in which only PACs produce filopodia (Fig. 8D-D′). In addition, we examined stage 8 follicles when all PFCs are in tight contact with the oocyte and express *ena*. At this stage, we found that all PFCs produce filopodia penetrating the oocyte, as assessed with mCD8::GFP marker expression (Supplementary Fig. 6E). Strikingly, at stage 10, we found that *ena* ectopic expression even in FC clones at a great distance from PCs (PC + 12) and, thus, not considered PFCs, was sufficient to induce filopodia formation and to maintain the tight membrane contact to the oocyte (Fig. 8E-E″). By contrast, control cells adjacent to these clones and closer to PCs were not able to elaborate filopodia nor maintain

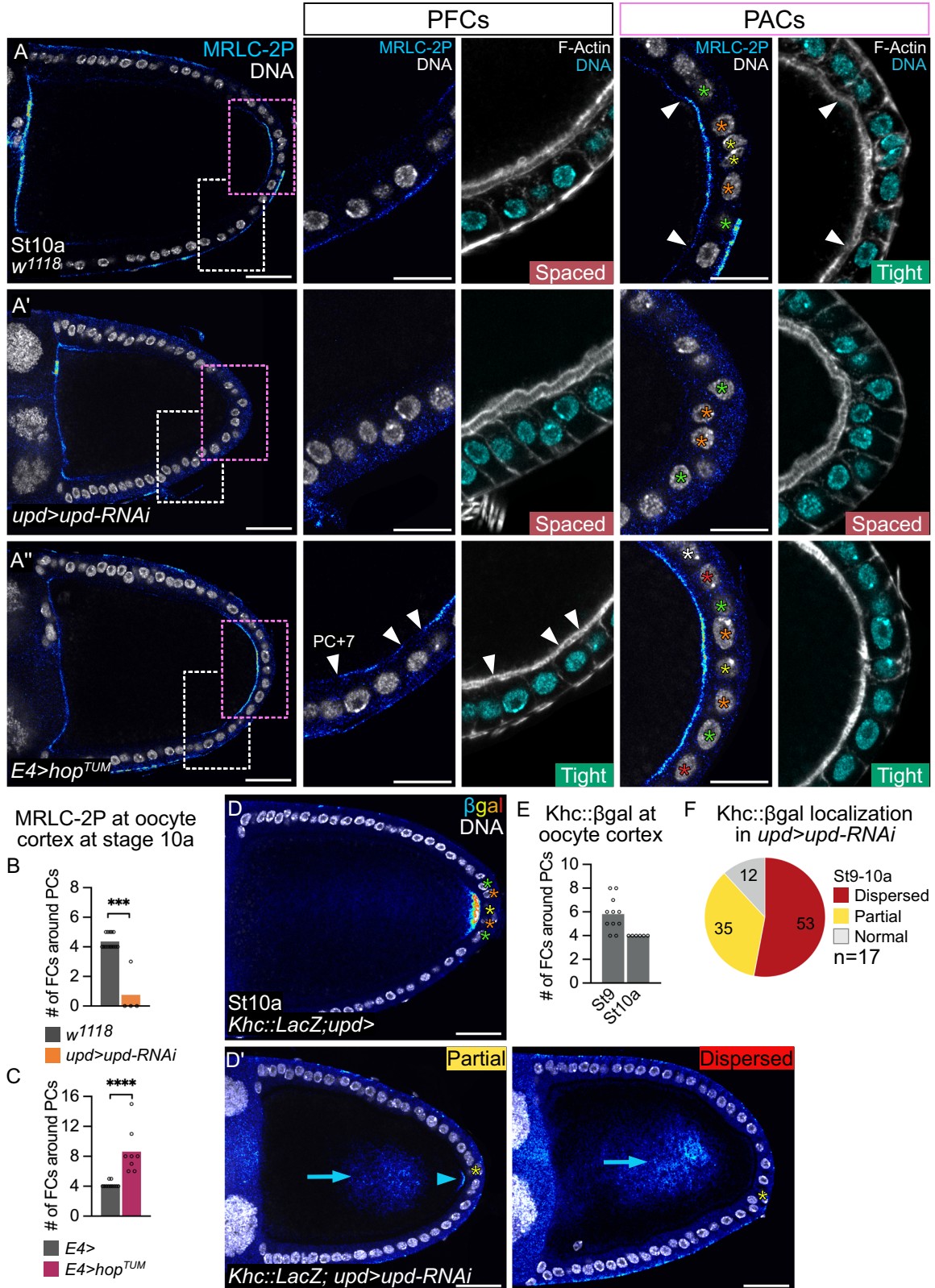

tight membrane contact with the oocyte. Together these results indicate that *ena* is sufficient, cell autonomously, not only to provide FCs with the ability to produce filopodia, but also to maintain the FC-oocyte interface tight, which is necessary for *oskar* mRNA anchoring.

Next, given the correlation we identified between the tight PAC-oocyte membranes and the activation of Myosin II in the oocyte, we tested whether the region with di-phosphorylated MRLC was also

modified upon *ena* gain-of-function. Indeed, we found that *ena* ectopic expression leads to an increase in the size of the MRLC-2P domain at the oocyte posterior cortex, which correlates precisely with the enlarged zone where oocyte and PFC membranes are maintained in tight contact (Supplementary Fig. 9A–B, white arrowheads). Finally, we sought to determine whether *ena* ectopic expression in PFCs has significant consequences on the future embryo, similarly to what occurs

**Fig. 4 | JAK-STAT signaling in PFCs is necessary and sufficient for MRLC di-phosphorylation and microtubule network polarity in the oocyte.** Posterior is to the right. Yellow, orange, green, red and white stars indicate PCs, PC + 1, +2, +3 and +4 rows respectively. **A,A',A"** Control follicle (*w*^III8^) and follicles expressing *upd-RNAi* in PCs under the control of *upd-Gal4* or *hop*^TUM^ in PFCs under the control of *E4-Gal4* are stained for MRLC-2P (royal filter), F-Actin (gray, phalloidin), DNA (gray or cyan, DAPI). PACs (pink dotted rectangles) and adjacent PFCs (white dotted rectangles) are magnified. "Tight" and "Spaced" refer to the perivitelline space. White arrowheads point to the correlation between the end of the MRLC-2P signal and the separation of oocyte and PFC membranes. **B, C** Quantifications of the MRLC-2P region size in number of facing FCs around PCs in control follicles, *upd* knocked down or *hop*^TUM^ follicles. **D,D'** Accumulation of *Khc::LacZ* to assess microtubule network polarity in control follicles and in follicles expressing *upd-RNAi* in PCs under the control of *upd-Gal4*, stained for β-galactosidase (βgal, royal filter), and DNA (gray, DAPI). **D'** Max z-projections of 2 consecutive 1 μm confocal sections. Phenotypes are classified from "Partial" (both smaller Khc::βgal accumulation, facing <4 PFCs, blue arrowhead, and dispersed in cytoplasm, blue arrow) to "Dispersed" (absent from cortex). **E** Quantification of Khc::βgal accumulation size as the number of facing FCs around PCs in control. **F** Khc::βgal phenotype categories in *Khc::LacZ;upd>upd-RNAi* stage 9 and 10a follicles. Those defects were never observed in control follicles. St: stage. #: number. Scale: 30 μm except magnifications 15 μm. Statistical tests: two-sided Mann–Whitney. Information about statistics and reproducibility is provided in Supplementary Tables 14–15. Source data are provided as a Source Data file.

when JAK-STAT is activated ectopically in all PFCs. To this end, we counted Vasa-positive cells in fertilized embryos arising from ovarian follicles expressing *ena* in all PFCs with the *E4-Gal4* driver. We found that these embryos have a significantly higher number of PGCs than control embryos, indicating that the ectopic expression of *ena* in PFCs is sufficient to produce an excess of PGCs in the future embryo (Fig. 8F–G).

Altogether, our results support a model whereby *ena* expression in PACs, in response to JAK-STAT signaling, allows the formation of filopodia that extend into the oocyte, keeping the oocyte and PAC membranes in tight contact. This would be required for maintaining MRLC di-phosphorylation in the oocyte, which is necessary to focus *oskar* mRNA polarized transport. This process, essential for recruitment of the proper amount of *oskar* mRNA during oogenesis, will subsequently determine the appropriate number of PGCs in the embryo.

## Discussion

Up until now, overexpression of *oskar* in the oocyte was the only known condition sufficient to produce ectopic PGCs in the early embryo[4,5]. Restricting the amount of posterior *oskar* mRNA during oogenesis is therefore crucial to determine PGC number. However, how this restriction is achieved has not been addressed so far. Our finding that restricted JAK-STAT signaling in the somatic follicular epithelium is required to limit the number of PGCs formed in the embryo led us to identify a subpopulation of follicle cells, the PACs, that had never been described before. PACs are necessary and sufficient to define a region of the posterior oocyte cortex for transporting *oskar* mRNA via a polarized cytoskeleton. We reveal that restriction of the size of this region in the oocyte is ensured through opposite actions of the Grk/EGFR and the JAK-STAT signaling pathways on *shg* expression in the follicular epithelium. We propose that this crosstalk occurs as follows: first, Grk is released from the oocyte at stage 6, thereby activating EGFR signaling in the adjacent follicular epithelium, which inhibits *shg* expression in PFCs around the oocyte. Consequently, JAK-STAT activity decreases at the posterior pole of follicles[33]. Nevertheless, a few PFCs, the PACs, the cells closest to the source of the Upd ligand released from PCs, maintain high JAK-STAT activity. While the rest of the PFCs continue to display low *shg* expression, JAK-STAT signaling activates the expression of *shg* in PACs. Similarly, *shg* is a JAK-STAT signaling target in another specific FC subpopulation, the border cells, in which it interacts genetically with *Stat92E* during their migration at stage 9[58–61]. The positive feedback loop between JAK-STAT signaling and E-Cadherin, which determines a robust number of PACs (about 20), may contribute to define the sharp boundary of high JAK-STAT activity required to localize the correct amount of *oskar* mRNA at the posterior oocyte cortex, thereby determining the appropriate number of PGCs in the future embryo.

Around 30 years ago, it was postulated that a trigger emanating from differentiated PFCs at stage 6 in response to Grk/EGFR signaling, signals back to the oocyte to reorganize the microtubule network essential for establishing anterior-posterior polarity[10,11,14,15]. At first,

hypotheses about the back signaling event evoked a chemical signal diffusing from PFCs. However, mosaic PFCs mutant for various genes can affect the localization of *oskar* mRNA in front of mutant cells, while maintaining it in front of adjacent wild-type cells, suggesting that the trigger coming from PFCs could be physically localized[12,16,17,62]. Our findings demonstrate that the oocyte, which is in tight contact with all PFCs at stage 6 when the back signaling is emitted, subsequently restricts its contact only to the PACs between stages 8 and 10, in response to JAK-STAT signaling. We found that this specific contact correlates perfectly with the region of myosin activation at the oocyte cortex, which is necessary for *oskar* mRNA posterior transport. Recently, it was shown that separating oocyte and PFC membranes manually at stage 9 leads to a loss of *oskar* mRNA anchoring[30]. We found that decreasing JAK-STAT signaling in the follicular epithelium leads to early separation of the PAC-oocyte membranes and affects *oskar* mRNA anchoring. Together, these results provide strong evidence that physical contact between the follicular epithelium and the oocyte is essential at stages 9 and 10 of oogenesis for *oskar* mRNA localization. Importantly, we performed several gains of functions (*hop*^TUM^, *shg*, *ena*) using *E4-Gal4*, a somatic driver that is active as of late stage 8, when *oskar* mRNA is already localized at the oocyte posterior cortex. In all cases, we found that the number of PACs and the area of tight contact between PACs and the oocyte, as well as Myosin II activity and the zone of *oskar* mRNA anchoring at the posterior cortex of the oocyte were all increased. Together, our results indicate that the organization of the microtubule cytoskeleton is still under the control of the follicular epithelium much later after microtubule polarity in favor of *oskar* polarized transport is established. The role of the follicular epithelium in polarizing the oocyte is therefore not limited to the back signaling sent at stage 6 after PFC differentiation. We rather believe that *oskar* mRNA posterior transport requires continued instruction from the posterior follicular epithelium to the oocyte until late stage 10, when PAC and oocyte membranes eventually separate. Our findings therefore challenge the established model that has persisted for several decades (reviewed in St Johnston, 2023).

We propose that JAK-STAT signaling acts reiteratively in the follicular epithelium to orchestrate the restriction of *oskar* mRNA to the posterior oocyte cortex. Our results reveal that JAK-STAT signaling within the posterior follicular epithelium is necessary for differentiation of two PFC subpopulations at different stages. First, JAK-STAT activity determines PFC fate at stage 6[12]. Then, the size of the JAK-STAT activity gradient is restricted between stages 7 and 9[33], leading to the differentiation of a robust number of PACs. Whereas PFCs represent an average of 200 cells, PACs, located at the center of the PFC domain, differentiate later and are 10 times less numerous, with an average of 20 cells. Early induction of PFC differentiation occurs when JAK-STAT signaling is active in a broad domain of the posterior follicular epithelium, while, later, restricted JAK-STAT signaling leads to differentiation of a limited number of PACs from among PFCs. Interestingly, our findings are supported by a single-cell transcriptomic analysis of the ovary, which identified two cell populations at posterior follicles at

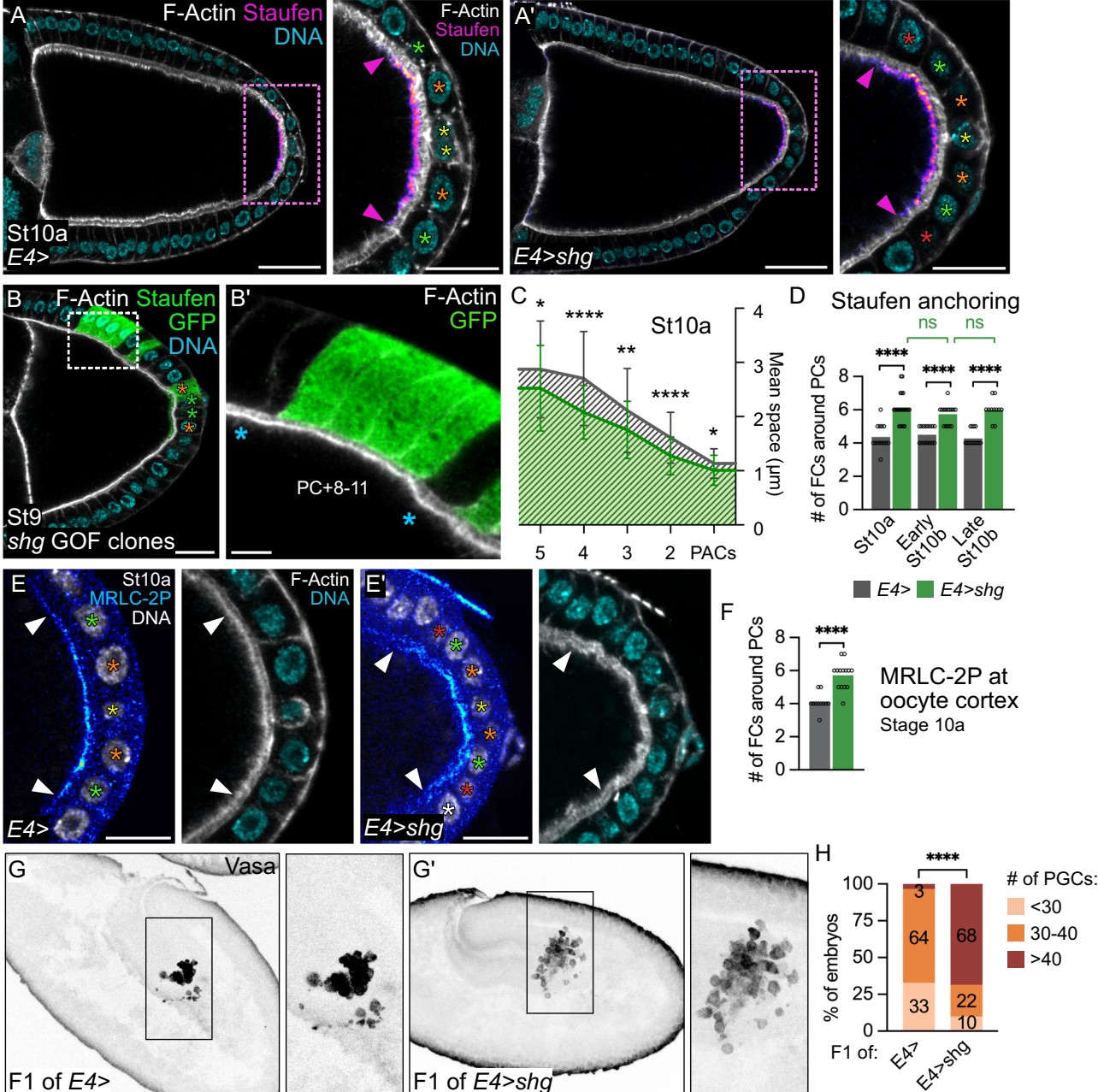

**Fig. 5 | E-Cadherin is sufficient to maintain PAC-oocyte membranes tight for *oskar* mRNA anchoring in the oocyte and to produce extra PGCs in the embryo.** Posterior is to the right. Follicles are stained for F-Actin (gray, phalloidin), DNA (cyan, DAPI), Staufen (fire filter), GFP (green) and MRLC-2P (royal filter). Yellow, orange, green, red and white stars indicate PCs, PC +1, +2, +3 and +4 rows respectively. **A,A'** Stage 10a control follicles (*E4*-Gal4/+) and follicles expressing *shg* in PFCs under the control of *E4*-Gal4. PACs (pink dotted rectangles) are magnified. *oskar* mRNA anchoring zone boundary is indicated by pink arrowheads. **B** Stage 9 follicle with mosaic clones (marked with GFP, green) expressing *shg* ectopically outside of the PFC domain. White dotted square indicates the magnified zone. Blue stars highlight the wider perivitelline space on each side of the clone. Phenotypes observed in 91% of cases. Quantifications (**C**) of the perivitelline space and (**D**) of the Staufen anchoring zone size as the number of FCs around PCs facing Staufen signal.

In (**C**), data are presented as mean values ± SD. **E,E'** PAC magnifications. White arrowheads highlight MRLC-2P signal boundary, correlating with a tight perivitelline space. **F** Quantification of the MRLC-2P region size in number of FCs around PCs facing MRLC-2P signal. **G,G'** Stage 9 embryos arising from control follicles and *E4>shg* follicles. Max z-projection showing Vasa (black) to identify PGCs. **H** Quantification of PGC number in stage 5–12 embryos. Numbers within bars indicate the percentage of embryos in each category. St: stage. #: number. Scale (**A, B**): 30 μm for (**A–E'**) and 50 μm for (**G–G'**) except magnifications 5 μm for (**B'**), 15 μm for (**A,A',E,E'**) and 30 μm for (**G–G'**). Statistical tests: two-sided unpaired *t* test for (**C**); two-sided Mann–Whitney for (**D, F**); two-sided Fisher's exact test for (**H**). Information about statistics and reproducibility is provided in Supplementary Tables 16–20. Source data are provided as a Source Data file.

stage 8, indicating that PFCs do not form a homogenous cell population[56]. Similarly to what occurs in the anterior region of follicles at stage 8 where JAK-STAT signaling defines four cell types (border, stretched, centripetal and mainbody cells)[12], we found that JAK-STAT signaling induces several fates at the posterior poles of follicles to

ensure the correct restriction of *oskar* mRNA transport and anchoring to a well-defined region.

Our work has revealed the existence of actin protrusions penetrating the oocyte that contain Ena, a protein involved in filopodia formation. These protrusions emanate from all PFCs at stage 8, and

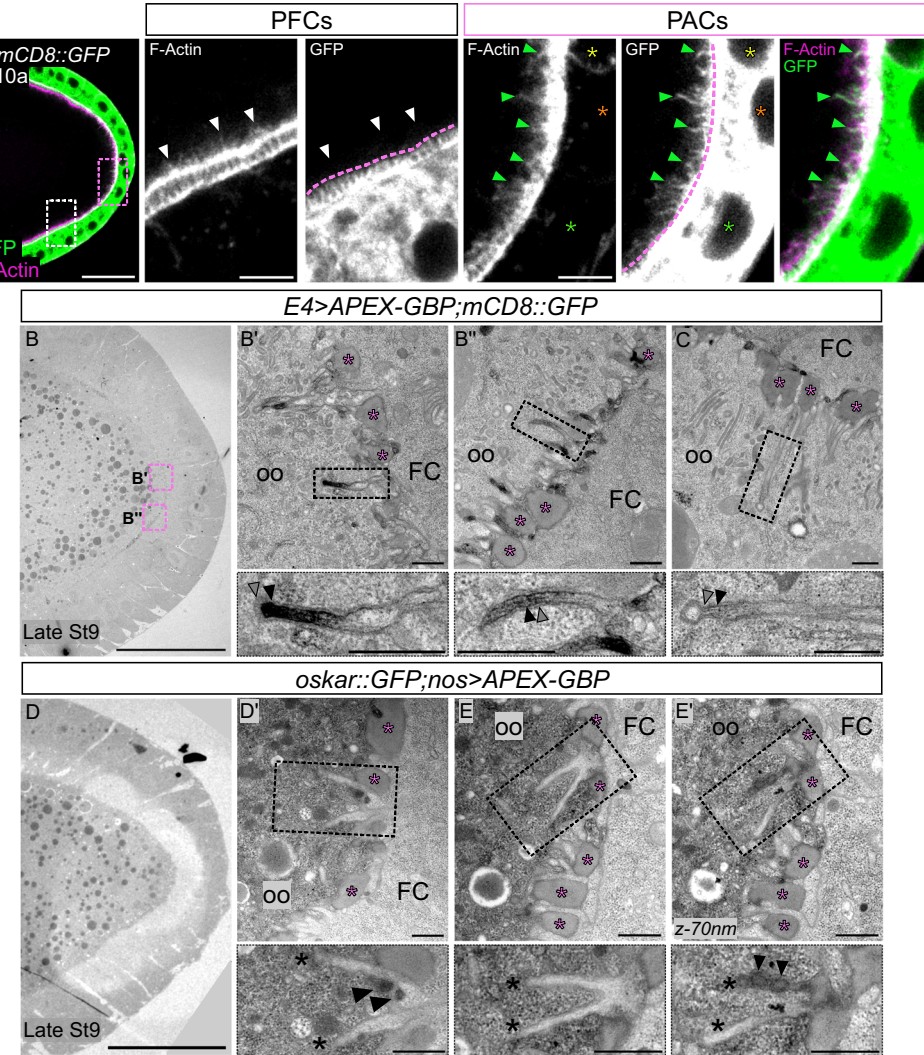

**Fig. 6 | The PACs form filopodia penetrating the oocyte where they can be found in contact with particles containing Oskar.** Posterior is to the right. In (**B**–**E**), pink stars indicate the perivitelline space. **A** FC/oocyte boundary in the region of PACs (pink dotted rectangle) and adjacent PFCs (white dotted rectangle) are magnified. Stage 10a follicle expressing *mCD8::GFP* in FCs under the control of *tj-Gal4*. In merges, GFP is in green and F-Actin in magenta (phalloidin). Yellow, orange and green stars indicate PCs, PC + 1, and +2 rows respectively. Green and white arrowheads point to GFP + /F-Actin+ filopodia and GFP-/F-Actin+ protrusions respectively. Pink dotted lines delimit the oocyte cortex. **B**, **C** EM images of late stage 9 follicles expressing *mCD8-GFP* and *APEX-GBP* in PFCs under the control of *E4-Gal4*, allowing the detection of GFP through an electron-dense precipitate. **C** Is a

magnification from another follicle than the one presented in (**B**). GFP+ protrusions are magnified (black dotted rectangles). GFP- membranes correspond to the oocyte (gray arrowheads), while underlying GFP+ membranes correspond to PACs (black arrowheads). **D**, **E** EM images of late stage 9 follicle expressing both *APEX-GBP* in germ cells under the control of *nanos-Gal4* (*nos*) and *oskar::GFP*. PAC protrusions are magnified (black dotted rectangles). Section (**D'**) and two consecutive sections 70 nm apart (**E-E'**), showing filopodia (black stars) in contact with Oskar-GFP particles (black arrowheads). St stage, oo oocyte, FC follicle cell. Scale: 30 μm except magnifications 5 μm for (**A**), and 500 nm for (**B'-C,D'-E'**). Information about reproducibility is provided in Supplementary Table 21.

later restrict to the PACs. Interestingly, *ena* is sufficient to provide FCs with the ability of elaborating filopodia, and to maintain the tight interface between PACs and the oocyte, even outside of the PFC domain, suggesting that filopodia could be directly involved in adhesion between the two tissues. Recently, membrane protrusions at the interface between the oocyte and FCs have been observed exclusively at the most posterior end of follicles using electron microscopy, as early as in the germarium and up to stage 7[63]. Posterior FCs therefore seem to share a specific interface with the oocyte throughout oogenesis. These early protrusions extend from both the oocyte and follicle cells in such a way that they interdigitate at the interface between the two tissues and could contribute to maintain the oocyte anchored at the posterior pole of follicles. This is reminiscent of the interdigitated actin-based short protrusions described at the interface

between neighboring epithelial cells in culture, which are thought to increase cell-cell contact by increasing the surface area of the cell where E-Cadherin is found[64]. The authors propose that this could reinforce E-Cadherin-based adhesion as compared to the presence of only the relatively weak E-Cadherin homophilic bonds. Actin filopodia extending into neighboring cells have also been described in other *Drosophila* developmental contexts. For instance, filopodia with E-Cadherin at the plasma membrane emanate basally in FCs and penetrate neighboring FC cortexes, thereby functioning as cell-cell anchoring sites required for follicle elongation[65]. In the germarium, cytonemes and cytosensors, which are also actin-based cellular extensions, are used to connect different cell types to modulate signaling for germline stem cell maintenance[66,67]. PAC filopodia could therefore play such roles as, transmitting molecules to the oocyte,

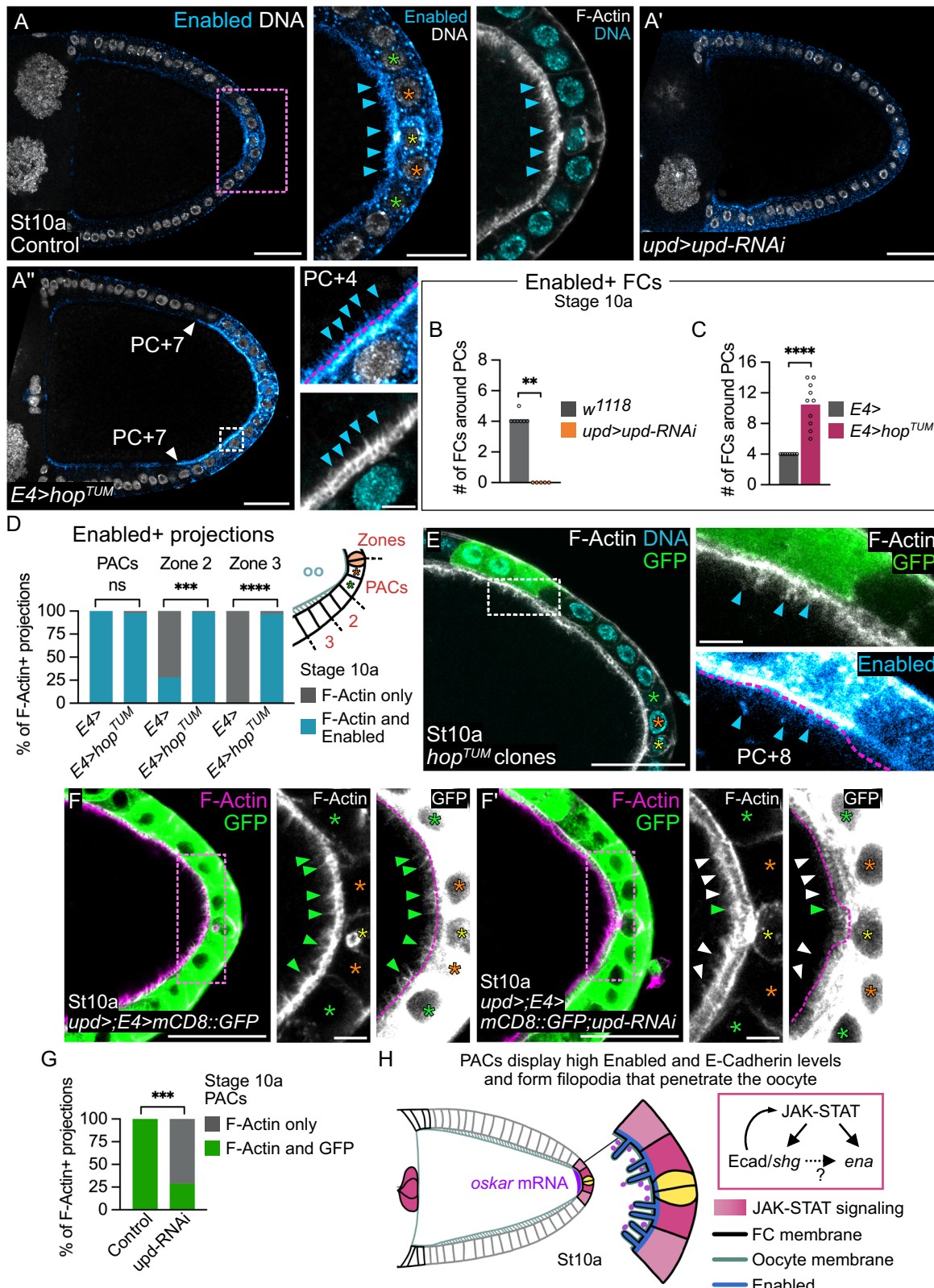

**D** Enabled+ projections

**H** PACs display high Enabled and E-Cadherin levels and form filopodia that penetrate the oocyte

activating Myosin II or maintaining adhesion between PACs and the oocyte at a stage where adherens junctions have been lost[63]. Alternatively, PAC filopodia penetrating the oocyte could provide a large cortical surface area within this specific region of the oocyte for anchoring of sufficient *oskar* mRNA, as supported by our finding that *oskar* mRNA particles can be found in contact with filopodia. By affording surface asperities, filopodia may also physically protect *oskar*

mRNA from cytoplasmic flows in the oocyte that could otherwise detach it from the actin cortex.

In addition to *ena*, we found that *shg*, like JAK-STAT signaling activity, is also sufficient to maintain a tight interface between the PACs and the oocyte, even outside the PFC domain, and to determine the size of *oskar* mRNA anchoring zone at the posterior pole of follicles. We hypothesize that E-Cadherin could have multiple roles in PACs, but

**Fig. 7 | The PACs display high levels of Enabled, a target of the JAK-STAT pathway that is sufficient to elaborate filopodia.** Posterior is to the right. Yellow, orange and green stars indicate PCs, PC + 1, and +2 rows respectively. In (**A**,**A′**,**A″**,**E**), blue arrowheads point to Enabled + /F-Actin+ filopodia. **A-A″** F-Actin (gray, phalloidin), DNA (gray or cyan, DAPI) and Enabled (cyan hot filter) staining in control follicles (*E4/+*), and follicles expressing *upd-RNAi* in PCs under the control of *upd-Gal4* or *hop^TUM* in PFCs under the control of *E4-Gal4*. In (A,A″), PACs (pink dotted rectangle) and PFC/oocyte interface (white dotted rectangle) are magnified. In (**A″**), white arrowheads point to the increased limit of Enabled+ FCs at PC + 7 position. **B**, **C** Quantifications of the number of Enabled+ FCs around PCs in stage 10a follicles. **D** Quantification of the proportion of F-Actin projections positive and negative for Enabled in 3 different zones in stage 10a follicles. **E** Follicle with a mosaic clone (marked with GFP, green) expressing *hop^TUM* outside the PFC domain, stained for F-Actin (gray, phalloidin) and DNA (cyan, DAPI). FC/oocyte interface at clone boundary (white dotted rectangle) is magnified. Phenotypes observed in 100% of cases. **F,F′** *upd-Gal4* and *E4-Gal4* drive expression of *mCD8::GFP* alone or together

with *upd-RNAi* in both PCs and PFCs. In merges, GFP is in green and F-Actin in magenta (phalloidin). The PAC/oocyte interfaces (pink dotted rectangle) are magnified. Green and white arrowheads point to filopodia positive for both GFP and F-actin and only F-Actin respectively. Pink dotted lines delimit the oocyte cortex. **G** Quantification of the proportion of F-Actin projections positive for GFP in stage 10a follicles in the PAC region. **H** Schematic representation of stage 10a follicle posterior region highlighting filopodia organization at PAC apical membrane. Filopodia penetrate the oocyte by deforming its plasma membrane, contacting *oskar* mRNA particles. In response to JAK-STAT signaling, PACs display high levels of Enabled, a protein involved in filopodia formation. St: stage. #: number. Scale: 30 μm except magnifications 15 μm for (**A**) and 5 μm for (**A″**,**E**,**F**,**F′**). Statistical tests: two-sided Mann–Whitney for (**B**, **C**); two-sided Fisher's exact test on mean number of projections for (**D**, **G**). Information about statistics and reproducibility is provided in Supplementary Tables 22–25. Source data are provided as a Source Data file.

is unlikely to have a classical junctional role at stages 9 and 10, as adherens junctions between PFCs and the oocyte are not observed after stage 7[63]. Since Cadherins are also involved in maintaining apico-basal polarity in epithelial cells, and since the apical surface in particular is compromised in Cadherin loss of function contexts[68,69], E-Cadherin could be required for PAC apico-basal polarity, which is likely to be important for the localization of filopodia at the PAC apical surface to penetrate the oocyte. Alternatively E-Cadherin could participate more directly in filopodia formation. For instance, *shg* is required for the formation and maintenance of invasive extensions during border cell migration[58]. E-Cadherin can also be present within filopodia themselves. During embryonic dorsal closure, *sisyphus* is required to transport cargo along filopodia, including E-Cadherin[70]. Another possibility is that E-Cadherin in PACs could be involved in orienting filopodia towards the oocyte. This is the case in nurse cells in which actin cables and E-Cadherin are interspersed, which is essential for correct orientation of actin cables and thereby for nuclear positioning[71].

Remarkably, follicle cells of a wide range of species (mouse, human, primates, cow, starfish, frog, zebrafish) elaborate thin cellular extensions that penetrate the extracellular coat to reach the membrane of the oocyte[72]. In mammals, transzonal projections are present between follicle cells and the oocyte. Although in most cases the projections terminate at the surface of the oocyte, in human, for example, some have been observed to penetrate deeply into the interior of the oocyte[73]. Hence, the formation of long protrusions connecting follicle cells and the oocyte seems to be evolutionarily conserved.

## Methods
### Fly stocks and genetics
The following stocks were used: *10XSTAT92E-GFP*[39], *shg-LacZ* (gift from JP. Vincent), *Khc::LacZ*[46], *UAS-hop^{TUM}*[74], *grk^{2E12}*, *Df(2 L)^{ED629}* (Gifts from V. Mirouse), *traffic-jam-Gal4* (DGGR, Kyoto), *fru-Gal4*[75], *E4-Gal4*[76], *upd-Gal4*[77], *UAS-GBP-APEX2*[54] and *oskar::GFP* (gift from A. Ephrussi). RNAi strains *UAS-upd-RNAi* (P{GD1158}v3282), *UAS-EGFR-RNAi* (P{KK100051} VIE-260B) are from VDRC. *w^{1118}*, *UAS-mCD8::GFP*, *hsFLP;tub < CD2< Gal4,UAS-GFP*, *nanos-Gal4VP16*, *UAS-ena::mcherry*, *UAS-shg.R*, *UAS-shg-RNAi* (P{TRiP.GL00646}attP40), *UAS-EGFR-RNAi* (P{TRiP.JF01084} attP2), *pnt-LacZ* (P{PZ}pnt[07825]) and *UAS-LacZ* (P{UAS-lacZ.NZ}J312) are from BDSC. Ethical approval is not required for *Drosophila* lines. *UAS-LacZ* transgenes were used to keep Gal4 to UAS ratios equal in control experiments with two UAS constructs. *10XSTAT92E-GFP, shg-LacZ* (homozygous only) and *pnt-LacZ* were maintained at 25 °C. All crosses using the *Gal4-UAS* system were performed at 25 °C until transfer of pharate pupae to 29 °C. Females were dissected at 3–4 days of age. Flip-out mosaic clones were obtained in pharate pupae or one-day-old adults raised at 25 °C by inducing 1 h heat-shocks at 37 °C on 3

consecutive days and females were dissected on the fourth day. When the *E4-Gal4* driver was used, since transgene expression is driven in all PFCs including PACs, genes endogenously expressed in PACs become overexpressed in PACs while ectopically expressed in other PFCs. Since, in gain of function experiments we analyzed effects outside of the PACs, we used the term ectopic for simplicity. For experiments on embryos, females of interest were crossed with control males and kept at 29 °C until embryo collection and fixation.

### Ovary immunostaining and native fluorescence
Ovaries were dissected in PBS and fixed for 15 minutes in 4% formaldehyde in PBS. After permeabilization in PBST (PBS-Triton 0.3%), ovaries were blocked in PBST with 2% BSA for 1 h. Incubation with primary antibodies was carried out on a shaker overnight at 4 °C. After several washes in PBST, ovaries were incubated 2 h with secondary antibodies, DAPI and phalloidin to label DNA and F-Actin, respectively. Ovarioles were washed in PBS, separated with fine needles and mounted on a slide in DABCO (Sigma). For native GFP observation of 10XSTAT92E-GFP females, ovaries were fixed, washed briefly in PBST, incubated with DAPI and phalloidin for 30 min with agitation and rinsed in PBS. Mounting and observations were carried out straightaway. Dilutions and references for antibodies are described in the Supplementary Method section of the Supplementary Data file.

### Embryo immunostaining
Eggs laid overnight by females of interest were bleached for 1 min, washed briefly in PBS and fixed in 273 μL PEMS, 33 μL 37% formaldehyde and 920 μL heptan for 30 min with agitation. Floating embryos were vortexed for 30 s in 500 μL heptan and 500 μL methanol. Sinking embryos were washed 3x in 1 mL methanol, 1x in PBS and 3x in PBST. Embryos were incubated in primary antibody solution overnight at 4 °C and in secondary antibody solution for 2 h at room temperature, each with agitation followed by washing 1x in PBS and 3x in PBST. Mounting was done in DABCO.

### Image processing and analysis
Confocal images were acquired with a Leica SP8 inverted microscope driven by Las-X software version 1.4.4, using 63x or 40x oil immersion lenses. All images are one confocal section passing through the middle of each follicle (i.e PCs visible) unless otherwise specified. For electron microscopy experiments, images were acquired with a transmission electron microscope Tecnai12 (Thermo Fisher Scientific). All images were processed with FIJI (version 2.14.0/1.54 f)[78] and annotated with Affinity Designer version 2.3 (Serif Europe Ltd, www.affinity.serif.com). Color filters range from one color, corresponding to low intensity, to another, corresponding to high intensity, with respectively purple to orange for "fire", blue to red for "royal", red to orange for "red hot", magenta to white for "magenta hot" and cyan to white for "cyan hot".

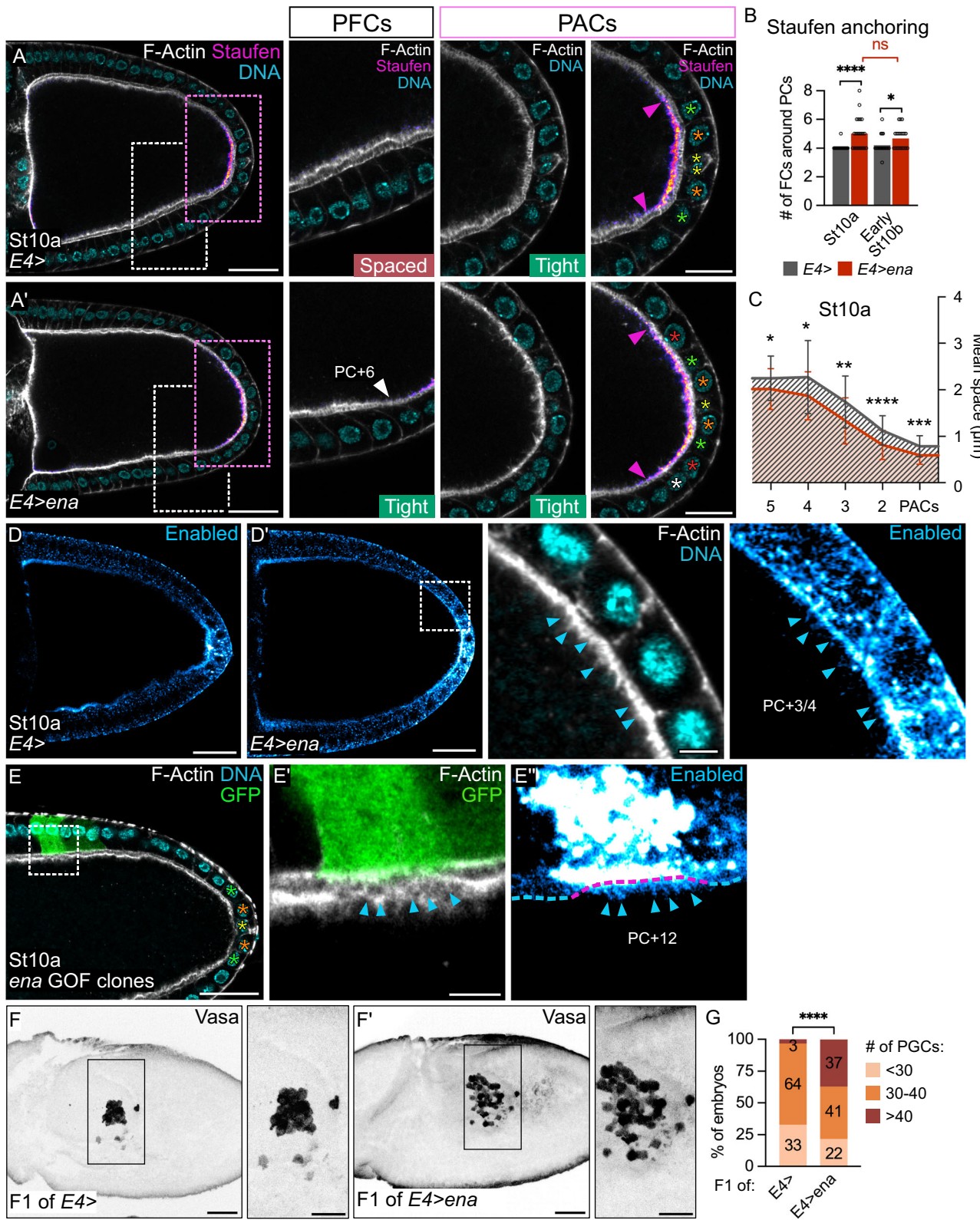

Quantifications on confocal stacks were performed using FIJI. To have consistent data regarding cell positions, similar images were used to determine the number of FCs (size of the *oskar* mRNA anchoring zone and β-galactosidase or GFP positive cells), the diameter of the anchoring zone and the space between oocyte et PAC membranes. In scatter dot plots, bars represent mean measurements and each dot corresponds to one follicle. For mean space measurements, the follicular epithelium is divided in five zones of 2 adjacent FCs, zone 1 being the PACs (as schematized in Fig. 3B). The area and length of the separation between the membranes in front of each zone was measured to calculate the mean space. For each follicle analyzed, measures were made on both PC sides. In mean space line graphs, each dot represents average measurements made at the corresponding zone on multiple posterior poles and lines are visual guides. For PGC quantification, Vasa-positive cells were counted on each section of confocal z-stacks spanning the entire embryo. All schemes were hand-drawn on

**Fig. 8 | *enabled* is sufficient to maintain PAC-oocyte membranes tight and to produce extra PGCs in embryos.** Posterior is to the right. Yellow, orange, green, red and white stars indicate PCs, PC +1, +2, +3 and +4 rows respectively. **A,A'** Control follicle (*E4/+*) and follicle expressing *ena* in PFCs under the control of *E4-Gal4* are stained for F-Actin (gray, phalloidin), DNA (cyan, DAPI) and Staufen (fire filter). PACs (pink dotted rectangles) and adjacent PFCs (white dotted rectangles) are magnified. "Tight" and "Spaced" refer to perivitelline space. Staufen anchoring zone boundary is indicated by pink arrowheads. In *E4>ena* follicle, the white arrowhead points to the increased region of tight PFC-oocyte contact. Quantifications of (**B**) the size of Staufen anchoring zone as the number of FCs around PCs facing Staufen signal and of (**C**) perivitelline space. In (**C**), data are presented as mean values ± SD. **D,D'** Control and *E4>ena* follicles stained for F-Actin (gray, phalloidin), DNA (cyan, DAPI) and Enabled (cyan hot filter). In (**D'**), a PFC/oocyte interface (white dotted rectangle) is magnified. **E-E"** Follicle with a mosaic clone

(marked with GFP, green) expressing *ena* outside of the PFC domain, stained for F-Actin (gray, phalloidin), DNA (cyan, DAPI) and Enabled (cyan hot filter). **E',E"** FC/oocyte interface in a clone region (white dotted rectangle) is magnified. Blue arrowheads point to Enabled + /F-Actin+ filopodia. Phenotypes observed in 29% of cases (*n* = 21). **F,F'** Stage 9 embryos arising from control (*E4-Gal4/+*) and *E4>ena* follicles. Max z-projection showing Vasa (black) to identify PGCs. **G** Quantification of PGC number in stage 5–12 embryos. Numbers within bars indicate the percentage of embryos in each category.St: stage. #: number. Scale: 30 μm for (**A**–**E'**) and 50 μm for (**F,F'**) except magnifications 15 μm for (**A,A'**), 5 μm for (**D',E'**) and 30 μm for (**F,F'**). Statistical tests: two-sided Mann–Whitney for (**B**); two-sided unpaired *t* test for (**C**); two-sided Fisher's exact test, light/dark orange versus brown categories, for (**G**). Information about statistics and reproducibility is provided in Supplementary Tables 26–29. Source data are provided as a Source Data file.

Procreate for iPad version 5.3.7 (Savage Interactive Pty. Ltd., www.procreate.com).

## Statistics and reproducibility

Detailed information about statistics and reproducibility are provided in a "Statistics and reproducibility" section of the Supplementary Data file, including exact *p* values and sample sizes. All statistical analyses were performed using GraphPad Prism version 10.1.2 for Mac (GraphPad Software, Boston, Massachusetts USA, www.graphpad.com). For comparison of means, the normality of the variables was checked and two-sided parametric (unpaired *t* test) or non-parametric (Mann Whitney) comparison tests were performed with a confidence interval of 95%. For contingency analysis, two-sided Fischer's exact tests were used with a confidence interval of 95%. On graphs, ns refers to non statistically significant differences and stars indicate statistically significant differences with the following code: $*p < 0.05$ $**p < 0.01$ $***p < 0.001$, and $****p < 0.0001$. No statistical method was used to predetermine sample size. No data were excluded from the analyses. The experiments were not randomized. The Investigators were not blinded to allocation during experiments and outcome assessment.

## Reporting summary

Further information on research design is available in the Nature Portfolio Reporting Summary linked to this article.

## Data availability

The data generated in this study are provided in the Supplementary Information and Source Data file. Source data are provided with this paper.

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

## Acknowledgements
We wish to thank Erika Bach, Jean-Paul Vincent, Anne Ephrussi, Vincent Mirouse, Véronique Brodu and Antoine Boivin for sharing fly stocks and Anne Ephrussi, Talila Volt, and Robert Edwin Ward for sharing antibodies. We are grateful to Fred Bernard for helpful discussions and Jean-René Huynh for valuable comments on the manuscript. We also thank Fly-base, BDSC (Bloomington *Drosophila* Stock Center), VDRC (Vienna *Drosophila* Resource Center), DGGR (Kyoto *Drosophila* Stock Center), DSHB (Developmental Studies Hybridoma Bank) for resources and the Imagerie-Gif core facility supported by the Agence Nationale de la Recherche (ANR-11-EQPX-0029/Morphoscope, ANR-10-INBS-04/FranceBioImaging; ANR-11-IDEX-0003-02/Saclay Plant Sciences) for confocal imaging and the ImagoSeine core facility of the Institut Jacques Monod member of IBiSA and the France-BioImaging (ANR-10-INBS-04) infrastructure for EM experiments. This work was supported by a 3 year-PhD fellowship obtained from the "Ministère de l'Enseignement Supérieur et de la Recherche" for CM, by the Center National de Recherche Scientifique (CNRS), Paris-Saclay and Versailles St-Quentin-en-Yvelines Universities, by a grant from the "Ligue contre le cancer" for laboratory expenses (AMP), and by a grant from the "Fondation ARC pour la recherche sur le cancer" for a year of salary expenses for CM.

## Author contributions
CM performed the vast majority of experiments and data analyses. SN, FC and MM performed some experiments. SC conducted the EM experiments and analysis. AMP, JM and AG provided material, infrastructure and helped with discussions on the project. AMP, JM and MM obtained the financial support. MM and CM conceptualized, designed the research study and wrote the manuscript. MM supervised the project. All authors critically read and approved the manuscript.

## Competing interests
The authors declare no competing interest.
