## [Peer Review File · Nature Communications]

REVIEWER COMMENTS

Reviewer #1 (Remarks to the Author):

The manuscript proposed by C. Mallart et al studies a developmental mechanism that takes place during *Drosophila* oogenesis. The determination of the anteroposterior axis of the oocyte is a textbook subject because it involves knowing where, when and how the axis of the fly embryo originates. Despite decades of study and numerous papers from many laboratories, how the posterior region is defined is not well understood. Although it is known that it involves a dialogue between the cortex of the oocyte and the adjacent follicular epithelial cells, the nature of this dialogue remains uncertain.

By combining genetics and quantitative imaging, this work provides important observations that shed light on what may be the mechanism for defining the posterior domain. The main observations are that 1) the precise size of the posterior domain of the oocyte is the consequence of a sub-differentiation of a cell population within the posterior follicular cell population 2) this sub-differentiation is the result of a complex interplay between EGF-R signaling, E-Cadherin expression and JAK-STAT pathway 3) this sub-differentiation is associated with morphological changes in the follicular cells that favour direct contact and surface exchange between the follicular cells and the oocyte membrane with the formation of filipodia originating from the follicular cells and giving an interdigitated appearance to the posterior region 4) modulation of the size of the cell population undergoing morphological changes is sufficient to extend the size of the posterior domain.

Although the exact nature of the signal that defines the posterior domain remains unknown, this work is clearly an important step towards its identification. The data are very convincing and will have an impact on the work of anyone working on fly oocyte polarity and more generally will interest everyone working on cell polarity. In addition, the interaction between Jak-STAT, E-Cad and EGF-R signaling and the putative cell-cell communication role of filopodia are of general interest to developmental and cell biologists.

The current manuscript could nevertheless be improved.

Major points :

1) Jak-STAT activity and determination of PACs.

While the JAK-STAT activity reporter is shown in a control situation, it is not shown in different contexts that are supposed to modify its profile. This is the case in Figure 1 under HopTum conditions. In these conditions, does the extension of the posterior domain correspond to the extension of the JAK-STAT positive domain or are not all the cells competent to become PACs (see point #4)? This is even more striking in figure 2, which ends with a diagram explaining how the Jak-STAT positive domain is defined, assuming that it is the same as PACs, when there is no data to directly support this conclusion. The same applies to Figure 3.

2) The need for filopodia to define the posterior domain

An obvious question is whether filopodia are necessary to define the posterior domain of the oocyte. The authors should provide the results of experiments testing this point. Note that a negative result would not diminish the interest of this work.

3) Relationship between adherens junction and filopodia

Authors show that extending the expression domain of Ecad or ena can increase posterior domain size. They should check whether they influence each other at the protein level or whether they act in parallel.

4) notion of sufficiency

The notion of sufficiency appears several times in the text (i.e. with ectopic expression of HopTum, Ecad, ena). However, all these cases are in the domain of posterior follicular cells. Could these gains of function influence the shape of the follicular cells, the space between the follicular cells and the oocytes or even the recruitment of Stau outside this domain? Flip-out clone experiments would help to answer this question. Once again, negative results would not reduce the interest of the article but would allow being more precise about this notion.

5) Match between text and figure

In many cases, the figures, main text and captions do not match up. For example :

- The title of Figure 2 (and the title of the text) does not correspond to what is shown in the figure. It is more about the control of Ecad by EGFR and Jak-STAT than the control of Jak-STAT by EGFR and Ecad.
- It is stated several times that the Jak-Stat domain restriction defines the size of the posterior domain. But it is not the restriction itself that defines the size of the domain.
- When the authors use the E4 driver, is the aim to obtain overexpression or ectopic expression? Authors talk about overexpression, which might be true, but the aim seems to be to obtain ectopic expression.

Minor points :

- Line #113: A reference for nos-Gal4 should be provided. Is it the Nos:Gal4VP16 fusion used by many labs or a different line?
- Line #115 (and many others) : Many fly stocks, such as RNAi lines, are identified by their stock centre number. This is not appropriate as these numbers can change or even disappear. Please, use proper name of the insertion.
- Line #207-208 : authors should better explain what they mean by “JAK-STAT activity decreases progressively “ and make the link to the so-called restriction if it corresponds to this.
- Line #219 : I do not see in the figure the supporting data for the quantification of the cell number leading to the average of 19.3 cells.
- Line #229 : “E-Cadherin, which we identified previously as a JAK-STAT signaling regulator”. Further explanation would be helpful.
- Line #342 : unformatted reference in the text.
- Line #427 : How the authors became interested in ena is something of a mystery.

- Line # 485 and elsewhere in the text : is it “boundary” rather than “boundaries” ? if I have understood correctly, it is a single border delimiting a circle centered on the posterior polar cells.

Reviewer #2 (Remarks to the Author):

This manuscript follows up the Pret group's previous observations that both E-cadherin and JAK/STAT signalling are down regulated in the posterior follicle cells after stage 6 to show that this down regulation is induced by Gurken/EGFR signalling from the oocyte. More importantly, they find that the posterior follicle cells within two cell diameters of the polar cells, the source of the JAK/STAT ligand, maintain E-cadherin levels and JAK/STAT signalling and adhere more closely to the underlying oocyte to define where oskar mRNA is localised and anchored. They go on to show that these most posterior follicle cells, which they name the posterior anchoring cells (PACs), also up-regulate Enabled and produce filopodia that extend into the oocyte, which is probably important for defining where oskar mRNA is anchored.

This work shows that JAK/STAT signalling acts as a morphogen to pattern the posterior of the follicle cell epithelium, just as it does at the anterior, but that this involves extensive cross-regulation with EGFR signalling and E-cadherin. More significantly, their results cast a new perspective on how the posterior follicle cells signal to polarise the Drosophila anterior-posterior axis. They show that only a small group of about 20 cells are responsible for this function and that their activity is continually required to maintain oskar mRNA at the posterior. This may explain why the elusive follicle cell signal has not been identified in over 20 years and raises the possibility that the signal is mechanical, rather than chemical. The results of Mallart et al. therefore represent a significant advance in the field. The experiments are well-designed with the appropriate controls and have been carefully quantified and their Figures are beautiful. I therefore strongly recommend that this manuscript be accepted by Nature communications without revision.

Reviewer #3 (Remarks to the Author):

In this thorough and well-designed study, the authors investigate the mechanism of oskar and stauferin localization to the posterior of the oocyte during Drosophila oogenesis. Through a series of experiments using genetics, fluorescence microscopy, and electron microscopy, the authors provide clear evidence that a subset of follicle cells, which they term posterior anchoring cells (PACs) direct oskar localization to a specific domain in the posterior of the oocyte. Specifically, gurken from the oocyte downregulates shg in all nearby follicle cells except those in close proximity to the posterior polar cells (the PACs). The PAC domain is defined by the extent of JAK-STAT signaling, which is activated by upd from polar cells. JAK-STAT upregulates shg in PACs, which facilitates a tighter association with the oocyte and helps to restrict myosin II activity to the PAC domain. This, in turn, patterns oskar localization. In addition, the authors provide evidence that JAK-STAT activates expression of the actin polymerization protein, enabled, which may induce the formation of protrusions from PACs into the oocyte, and that overexpression of enabled in posterior follicle cells expanded the domain of oskar localization. All together, the results provide strong support for the conclusions and I believe the study would be of interest to the field. I would recommend that the authors consider the following minor points before publication:

1. The study is framed as being motivated by an interest in understanding how oskar is patterned during oogenesis so that the proper number of PGCs are specified in the embryo but the authors do not revisit this phenotype with any subsequent mutants. Is PGC number altered by other genetic perturbations, such as ena or shg overexpression, as expected?
2. It would be informative to see the 10x STAT-GFP channel in the 3D projection shown in Fig. 1H'
3. In Fig. 5K, which pairwise comparisons are the ns and ** designations referring to?
4. Is overexpression of ena with E4-Gal4 sufficient to expand the region in which follicle cell filopodia projections into the oocyte? If so, can the authors speculate on how filopodia formation directs the patterning of oskar localization in the posterior of the oocyte?

Dear Reviewers,

We are pleased to submit the revised version of our manuscript. We would like to deeply thank you for your very positive and constructive comments on our manuscript, which helped us improving the quality of our work. We have now addressed all your comments and therefore several additional results have been included in the manuscript. In particular, our new results show that ectopic JAK-STAT activity, *shotgun* and *enabled* expression are each sufficient to maintain a tight contact between the follicular epithelium and the oocyte, even outside the domain of posterior follicle cells. Interestingly, the PACs elaborate filopodia penetrating the oocyte to maintain a tight contact with the oocyte, and we show now that filopodia formation is lost when JAK-STAT signaling is decreased. In addition, *enabled* is sufficient to provide follicle cells with the ability of elaborating those filopodia. By using electron microscopy, we also found filopodia in contact with Oskar-containing particles at the posterior oocyte cortex. Importantly, we show that ectopic expression of *shotgun* and *enabled* in the follicular epithelium are each sufficient to produce extra PGCs in developing embryos. Together, our new data reinforce our model showing the importance of determining a limited number of PACs in the posterior follicular epithelium to maintain a specific contact with the oocyte, which is required for the polarized transport of oskar mRNA and for the development of a correct number of PGCs in the future embryo.

Reviewer #1 (Remarks to the Author): The manuscript proposed by C. Mallart et al studies a developmental mechanism that takes place during *Drosophila* oogenesis. The determination of the anteroposterior axis of the oocyte is a textbook subject because it involves knowing where, when and how the axis of the fly embryo originates. Despite decades of study and numerous papers from many laboratories, how the posterior region is defined is not well understood. Although it is known that it involves a dialogue between the cortex of the oocyte and the adjacent follicular epithelial cells, the nature of this dialogue remains uncertain. By combining genetics and quantitative imaging, this work provides important observations that shed light on what may be the mechanism for defining the posterior domain.

The main observations are that

- 1) the precise size of the posterior domain of the oocyte is the consequence of a sub-differentiation of a cell population within the posterior follicular cell population
- 2) this sub-differentiation is the result of a complex interplay between EGF-R signaling, E-Cadherin expression and JAK-STAT pathway
- 3) this sub-differentiation is associated with morphological changes in the follicular cells that favour direct contact and surface exchange between the follicular cells and the oocyte membrane with the formation of filopodia originating from the follicular cells and giving an interdigitated appearance to the posterior region

4) modulation of the size of the cell population undergoing morphological changes is sufficient to extend the size of the posterior domain. Although the exact nature of the signal that defines the posterior domain remains unknown, this work is clearly an important step towards its identification. The data are very convincing and will have an impact on the work of anyone working on fly oocyte polarity and more generally will interest everyone working on cell polarity. In addition, the interaction between Jak-STAT, E-Cad and EGF-R signaling and the putative cell-cell communication role of filopodia are of general interest to developmental and cell biologists. The current manuscript could nevertheless be improved. We thank the reviewer for her/his positive comments on the quality of our experiments and for acknowledging the relevance of our findings. We have addressed all the mentioned shortcomings to improve our study as detailed below.

Major points:

1) Jak-STAT activity and determination of PACs While the JAK-STAT activity reporter is shown in a control situation, it is not shown in different contexts that are supposed to modify its profile. This is the case in Figure 1 under HopTum conditions. In these conditions, does the extension of the posterior domain correspond to the extension of the JAK-STAT positive domain or are not all the cells competent to become PACs (see point #4)? This is even more striking in figure 2, which ends with a diagram explaining how the Jak-STAT positive domain is defined, assuming that it is the same as PACs, when there is no data to directly support this conclusion. The same applies to Figure 3. To address this comment, we drove the expression of hoptum in PFCs with the ena-Gal4 driver in flies carrying the JAK-STAT activity reporter 10XSTAT92E-GFP. This leads to an extension of the JAK-STAT activity domain at the posterior pole of follicles, as now illustrated in Fig.1K'-L. In addition, in this gain of JAK-STAT activity follicles, we observed a correlation between JAK-STAT signaling, as assessed with the reporter, and the domain where follicle cells and the oocyte are in tight contact (Fig.3E). To strengthen the data on the correlation between the JAK-STAT activity domain and the oskar anchoring zone, we also added an image of a follicle expressing the 10XSTAT92E-GFP reporter stained for Stauf, further illustrating the correlation in control condition (Fig.1K). Finally, we showed in the first version of the manuscript that gain of JAK-STAT activity leads to an extension of the domain of follicle cells expressing ena. We are now adding a zoom and quantifications highlighting that these cells do elaborate filopodia, a characteristic of PACs (Fig.7A",D), and a new result showing that hopTUM clones located even outside the PFC domain are sufficient for follicle cells to elaborate Ena positive filopodia penetrating the oocyte (Fig.7E). Hence, these data further support that more PACs are formed upon hoptum expression in PFCs. Together, our new results demonstrate that the expression of hoptum in PFCs indeed translates into an extension of JAK-STAT activity in the posterior domain of follicles, and consequently in the acquisition of the PAC fate. Whether all FCs surrounding the oocyte are competent to become PACs is further discussed in our response to point 4.

2) The need for filopodia to define the posterior domain. An obvious question is whether filopodia are necessary to define the posterior domain of the oocyte. The authors should provide the results of experiments testing this point. Note that a negative result would not diminish the interest of this work. We agree that the question of the necessity of filopodia to define the posterior domain of the oocyte is very interesting. We tried to interfere with filopodia formation in different ways. We first expressed an RNAi against *ena* in the follicular epithelium with the *fruitless-Gal4* driver that is more active at follicle poles but this only led to a mild decrease in *Ena* accumulation at stage 9-10, as detected with the anti-*Ena* antibody. We thus used a stronger somatic cell driver, *traffic-jam-Gal4* to target *ena*-RNAi to follicle cells. In this case, the RNAi was efficient at decreasing *Ena* levels since we could not detect *Ena* protein in PACs anymore. However, actin positive protrusions were still present at the oocyte cortex although we were unsure of their origin since it was difficult to distinguish if they were PAC filopodia or oocyte cortical actin filaments. The identification of filopodia was complicated by the fact that we could not use *Ena* as a filopodia marker. We cannot exclude that filopodia were affected in this hypomorphic condition, but we could not observe any obvious filopodia phenotype. In addition, we did not detect any phenotype on the PAC/oocyte interface, nor on *oskar* mRNA localization. The absence of phenotype could indicate that *ena* is not directly required for these processes. Alternatively, it is possible that residual *Ena* activity remains in this hypomorphic context, although *Ena* is below detection levels, which would be sufficient for filopodia formation. Another possibility is that other redundant factors acting on filopodia formation are expressed in PACs and could compensate for the decrease in *Ena* translation. Hence, the result of this experiment is not very informative, we therefore decided not to include it in the revised version of the manuscript. To try and answer the reviewer question, we thus adopted an alternative strategy by generating somatic clones bearing an hypomorphic *ena* mutation (*ena210*). This mutant has a point mutation affecting *ena* function (Ahern-Djamali et al., 1998) and we found that it led to a decrease in *Ena* levels in clones. We examined carefully many cases but we didn't identify clear phenotypes. Since this set of experiments was not conclusive, we did not include it in the manuscript. It is known that filopodia formation is under the control of the Rho small GTPase *Cdc42*. As another way to alter filopodia formation, we thus knocked down *cdc42* by targeting an RNAi in FCs with the *fruitless-Gal4* or the *tj-Gal4* driver. We also expressed a dominant negative form of *Cdc42* in FCs. In all cases, we didn't detect any change in *Ena* levels nor localization, nor on *oskar* mRNA localization, nor on PAC/oocyte interface. These results indicate that either *cdc42* is not involved in filopodia formation in PACs, either the knockdown and dominant negative constructs were not sufficient to impede *cdc42* function in those cells. We therefore did not include this result in the manuscript. Our different attempts to affect filopodia by decreasing levels of genes involved in filopodia formation were not conclusive enough to shed light on the role of filopodia in the PACs. Nevertheless, we sought to analyze filopodia formation in a context of decreased JAK-STAT signaling. We expressed *mCD8::GFP* in PFCs with the *E4-Gal4* driver to label filopodia, in follicles also presenting decreased JAK-STAT activity. Under this condition, the number of GFP positive filopodia coming from the PACs and penetrating the oocyte is significantly lower than those of

control follicles. This new result presented in Fig.7F-G, indicates that JAK-STAT signaling is necessary for filopodia formation. Since JAK-STAT signaling decrease leads to loss of filopodia, widening of PAC-oocyte interface and loss of oskar mRNA anchoring, and that *ena* is sufficient to extend the domain of the tight PAC-oocyte interface and of oskar mRNA anchoring, it is very likely that filopodia are necessary for oskar mRNA posterior localization. Finally, in response to reviewer 3, we also added two new results linking *ena* expression, filopodia formation and the determination of the oocyte posterior domain. First, we show that ectopic *ena* expression in PFCs with the E4-Gal4 driver is sufficient to expand the domain of cells projecting filopodia in the oocyte (Fig.8D,D'). Second, we show that this leads to a significant increase in PGCs in embryos arising from these follicles, as does a gain of JAKSTAT function (Fig.8F-G). This shows that, similarly to JAK-STAT signaling, *ena* is sufficient, not only to maintain PAC and oocyte membrane in close contact, but also affects germline development of the future embryo. Altogether, we believe that our new results strengthen the functional significance of *ena*, and thus on filopodia formation, to define the posterior domain of the oocyte required for targeting oskar mRNA localization to determine PGC number in the future embryo.

3) Relationship between adherens junction and filopodia Authors show that extending the expression domain of *Ecad* or *ena* can increase posterior domain size. They should check whether they influence each other at the protein level or whether they act in parallel. To address this question, we drove the expression of *ena* in PFCs with the E4-Gal4 driver and examined E-Cadherin levels in the posterior follicular epithelium. Although, as expected, *Ena* levels were detected in a larger domain than in control follicles, we didn't observe any change in E-Cadherin levels in the corresponding region. This indicates that *ena* function downstream of JAK-STAT signaling in PACs is not affecting E-Cadherin levels, suggesting that *ena* and *shg* are two JAK-STAT targets acting in parallel. For the reverse experiment, we drove the expression of *shg* in PFCs and examined *Ena* levels in the posterior follicular epithelium. In this case, the posterior domain displaying high *Ena* levels was expanded compared to controls. Two interpretations can be stressed from this result. Either E-Cadherin regulates *Ena* protein levels positively, either the upregulation of *Ena* levels is an indirect consequence of an increase in JAK-STAT signaling activity. Indeed, we have previously shown that E-Cadherin is a positive regulator of JAK-STAT signaling in the follicular epithelium between stages 7 and 9. To investigate whether E-Cadherin was still acting positively on JAK-STAT signaling at stage 10 in the posterior follicular epithelium, we drove *shg* expression in PFCs with the E4-Gal4 driver in follicles bearing the JAK-STAT activity reporter. We observed that the posterior domain of JAK-STAT activity was increased in response to *shg* gain of function, indicating that *shg* still acts as a positive regulator of JAK-STAT signaling at this stage and in this region. In conclusion, although we cannot exclude a direct role of *shg* on *Ena* levels, it is possible that the increase in *Ena* levels observed in this condition results from the positive feedback loop taking place between JAKSTAT signaling and E-Cadherin. We have added these new results to Supplementary Fig.8.

4) notion of sufficiency The notion of sufficiency appears several times in the text (i.e. with ectopic expression of *HopTum*, *Ecad*, *ena*). However, all these cases are in the domain of

posterior follicular cells. Could these gains of function influence the shape of the follicular cells, the space between the follicular cells and the oocytes or even the recruitment of Stau outside this domain? Flipout clone experiments would help to answer this question. Once again, negative results would not reduce the interest of the article but would allow being more precise about this notion. We agree that the notion of sufficiency outside the PFC domain is an interesting question that we had not addressed in the previous version of the manuscript. Based on the expression of the PFC marker pointed, which we quantified in Supplementary Fig.3, we considered follicle cells located further than a 5-cell radius around polar cells to be outside the PFC domain (see scheme added in Supplementary Figure 3C'). We thus generated clones expressing hoptum, shg or ena using the flip-out system or taking advantage of the mosaicism of the E4-Gal4 driver (see Supplementary Fig.1), and examined the clones which formed outside the PFC domain, i.e. from PC+6 and beyond. Strikingly, in the majority of cases, expression of the three transgenes outside PFCs was sufficient to maintain follicle cell and oocyte membranes tighter than in adjacent non-clonal follicle cells, with a stronger effect for hoptum and shg (100%, n=7 and 91%, n=22 respectively) than ena (29%, n=21). Regarding the localization of Stau however, none of the transgene was sufficient to localize it ectopically. This is in accordance to our previous result illustrated in Fig.3A". Indeed, we observe there that the extended domain of PAC/oocyte tight membranes obtained upon JAK-STAT ectopic activity within the PFC domain, is larger (PC+7) than the extension of Stau localization (PC+3). In conclusion, JAK-STAT signaling, shg and ena are each sufficient to maintain follicle cell and oocyte membranes tight even outside the PFC domain, but are not sufficient to localize oskar mRNA ectopically. These results were added to Fig.3E, 5B, 7E and 8E. In addition, we found that hoptum and ena were also sufficient to provide follicle cells outside the PFC domain with the ability to elaborate filopodia penetrating the oocyte, which we added to Fig.7E and 8E, respectively. Hence, coming back to the first point raised by the referee, these new results indicate that follicle cells are competent to become PACs, since they display characteristic features of PACs, but this is not sufficient for oskar mRNA ectopic localization outside the PFC domain.

5) Match between text and figure In many cases, the figures, main text and captions do not match up. For example: -The title of Figure 2 (and the title of the text) does not correspond to what is shown in the figure. It is more about the control of E-Cad by EGFR and Jak-STAT than the control of Jak-STAT by EGFR and E-cad. We agree that, although ultimately EGFR signaling and E-Cadherin act on JAK-STAT signaling since a feedback loop is established, the title of Figure 2 did not reflect correctly what is shown in the figure. We thus changed it for "Opposite actions of EGFR and JAKSTAT signaling pathways restrict E-Cadherin levels to a subpopulation of PFCs" We changed the corresponding text section title for "Restricted JAK-STAT signaling in the posterior follicular epithelium results from EGFR signaling effect on shg expression". To reinforce data on the feedback loop existing between JAK-STAT signaling activity and shg expression, we also added a result showing that shg is sufficient for JAK-STAT activity in PFCs in Supplementary Fig.8E-F (see also point 3). - It is stated several times that the Jak-Stat domain restriction

defines the size of the posterior domain. But it is not the restriction itself that defines the size of the domain. Indeed, it is JAK-STAT signaling that defines the size of the posterior domain. Signaling restriction is important since expanding the size of the posterior domain has deleterious consequences, but we agree that it is not the restriction itself that defines the domain. We corrected the text accordingly in several places by either suppressing the word "restriction" or by replacing for instance "restriction of JAK-STAT signaling defines..." by "restricted JAK-STAT signaling defines..." where restricted qualifies JAK-STAT signaling. - When the authors use the E4 driver, is the aim to obtain overexpression or ectopic expression? Authors talk about overexpression, which might be true, but the aim seems to be to obtain ectopic expression. The difference between the two adjectives ectopic and overexpression is subtle in that case. Our aim was indeed to obtain ectopic expression, i.e. in PFCs outside the PAC domain. However, since the E4-Gal4 driver triggers transgene expression in all PFCs (and even beyond), including the PACs, genes that are already expressed in PACs are overexpressed there, while also ectopically expressed in other PFCs. In addition, since for instance *shg* and *ena* are highly expressed in PACs but have nevertheless a basal low expression profile in all follicle cells, we thought that the use of ectopic could be slightly incorrect. We agree with the referee that ultimately, ectopic is more appropriate than overexpression since we assess phenotypes outside the control PAC domain when performing these gain of function experiments. To resolve that issue, we have changed overexpression for ectopic in the text and clarified the use we make of ectopic in that particular context in the methods section. We also added a scheme in Supplementary Fig.3C showing the extent of the three domains of FCs that are found around the oocyte: PACs, PFCs and lateral FCs. Minor points : - Line #113: A reference for *nos*-Gal4 should be provided. Is it the *Nos:Gal4VP16* fusion used by many labs or a different line? It is indeed this line, we have clarified it in the text. - Line #115 (and many others) : Many fly stocks, such as RNAi lines, are identified by their stock centre number. This is not appropriate as these numbers can change or even disappear. Please, use proper name of the insertion. It is true that stock center are subject to changes, we have therefore changed them for insertion with their unique ID when appropriate. - Line #207-208 : authors should better explain what they mean by "JAK-STAT activity decreases progressively " and make the link to the so-called restriction if it corresponds to this. We clarified by changing this part of the sentence for the following one: "JAK-STAT activity progressively restricts between stage 7 and late stage 8 to a smaller number of PFCs". - Line #219 : I do not see in the figure the supporting data for the quantification of the cell number leading to the average of 19.3 cells. We added the graph corresponding to this quantification in Fig.1J. - Line #229 : "E-Cadherin, which we identified previously as a JAK-STAT signaling regulator". Further explanation would be helpful. We changed this sentence for the following for clarification: "We previously showed that E-Cadherin regulates JAK-STAT signaling positively and that both JAK-STAT activity and *shg* expression follow a similar dynamics during oogenesis, leading to their progressive restriction at posterior poles of follicles." - Line #342 : unformatted reference in the text. We corrected the format for this reference. - Line #427 : How the authors became interested in *ena* is something of a mystery. We searched for actin regulators in the list of genes expressed in the posterior-most follicle cells at late stage 8 from

the single-cell mRNA sequencing data which has been recently published (Slaidina et al., 2021) and found *ena* as a candidate. We added this explanation in the text. - Line # 485 and elsewhere in the text : is it “boundary” rather than “boundaries” ? If I have understood correctly, it is a single border delimiting a circle centered on the posterior polar cells. It is indeed a circle and thus boundary should be used as a singular. We have changed the text accordingly.

Reviewer #2 (Remarks to the Author): This manuscript follows up the Pret group's previous observations that both E-cadherin and JAK/STAT signalling are down regulated in the posterior follicle cells at stage 6 to show that this down regulation is induced by Gurken/EGFR signalling from the oocyte. More importantly, they find that the posterior follicle cells within two cell diameters of the polar cells, the source of the JAK/STAT ligand, maintain E-cadherin levels and JAK/STAT signalling and adhere more closely to the underlying oocyte to define where *oskar* mRNA is localised and anchored. They go on to show that these most posterior follicle cells, which they name the posterior anchoring cells (PACs), also up-regulate *Enabled* and produce filopodia that extend into the oocyte, which is probably important for defining where *oskar* mRNA is anchored. This work shows that JAK/STAT signalling acts as a morphogen to pattern the posterior of the follicle cell epithelium, just as it does at the anterior, but that this involves extensive crossregulation with EGFR signalling and E-cadherin. More significantly, their results cast a new perspective on how the posterior follicle cells signal to polarise the *Drosophila* anterior-posterior axis. They show that only a small group of about 20 cells are responsible for this function and that their activity is constitutively required to maintain *oskar* mRNA at the posterior. This may explain why the elusive follicle cell signal has not been identified in over 20 years and raises the possibility that the signal is mechanical, rather than chemical. The results of Mallart et al. therefore represent a significant advance in the field. The experiments are well-designed with the appropriate controls and have been carefully quantified and their Figures are beautiful. I therefore strongly recommend that this manuscript be accepted by Nature Communications without revision. We are extremely grateful to the reviewer for his/her strong support and for acknowledging that our work represents a significant advance in the field.

Reviewer #3 (Remarks to the Author): In this thorough and well-designed study, the authors investigate the mechanism of *oskar* and *staufen* localization to the posterior of the oocyte during *Drosophila* oogenesis. Through a series of experiments using genetics, fluorescence microscopy, and electron microscopy, the authors provide clear evidence that a subset of follicle cells, which they term posterior anchoring cells (PACs) direct *oskar* localization to a specific domain in the posterior of the oocyte. Specifically, *gurken* from the oocyte downregulates *shg* in all nearby follicle cells except those in close proximity to the posterior polar cells (the PACs). The PAC domain is defined by the extent of JAK-STAT signaling, which is activated by *upd* from polar cells. JAK-STAT upregulates *shg* in PACs, which facilitates a tighter association with the oocyte

and helps to restrict myosin II activity to the PAC domain. This, in turn, patterns oskar localization. In addition, the authors provide evidence that JAK-STAT activates expression of the actin polymerization protein, enabled, which may induce the formation of protrusions from PACs into the oocyte, and that overexpression of enabled in posterior follicle cells expanded the domain of oskar localization. All together, the results provide strong support for the conclusions and I believe the study would be of interest to the field. I would recommend that the authors consider the following minor points before publication: We thank the reviewer for her/his positive comments on the quality of our experiments and for acknowledging the relevance of our findings to the field. We have addressed the minor points mentioned by the reviewer to improve our study as detailed below.

1. The study is framed as being motivated by an interest in understanding how oskar is patterned during oogenesis so that the proper number of PGCs are specified in the embryo but the authors do not revisit this phenotype with any subsequent mutants. Is PGC number altered by other genetic perturbations, such as ena or shg overexpression, as expected? We thank the reviewer for suggesting this additional experiment. To address this question, we assessed the number of PGCs in embryos resulting from follicles presenting gain of functions for ena and shg in the follicular epithelium, driven by the E4-Gal4 driver. In both cases, we found the number of PGCs in embryos to be significantly increased. This new result, which we added to Fig.5G-H and 8F-G respectively, strengthens our findings that the expression of ena and shg in PACs is functionally significant since it regulates the number of PGCs of the future embryo.

2. It would be informative to see the 10x STAT-GFP channel in the 3D projection shown in Fig. 1H'. We have added the 10X STAT-GFP channel in the 3D projection, which is now Fig.1H'.

3. In Fig. 5K, which pairwise comparisons are the ns and ** designations referring to? Please note that due to new space constraints, after adding all the additional data in the revised version of the manuscript, this panel has now been moved to Supplementary Fig.7D. It was written in the legend that the “stars indicate statistically significant differences between upd>;E4>updRNAi;LacZ and upd>;E4>updRNAi;shg follicles”. To improve visual clarity, we changed the color font of ns and ** to brown, following the color code of the upd>;E4>updRNAi;shg genotype and we highlighted the two genetic conditions that were subject to statistical analysis in the figure panel itself.

4. Is overexpression of ena with E4-Gal4 sufficient to expand the region in which follicle cell filopodia projections into the oocyte? If so, can the authors speculate on how filopodia formation directs the patterning of oskar localization in the posterior of the oocyte? To verify that gain of ena indeed induces the formation of filopodia in a larger region of the posterior follicles, we drove ena expression with the E4-Gal4 driver as suggested by the referee. Staining with anti-Ena antibody allowed the detection of filopodia emanating from PFCs in a larger region than in control conditions. This result, confirming that ena is sufficient to expand the region in which follicle cell project filopodia into the oocyte, was added in Fig.8D,D'. Finally, and in response to reviewer 1, point 4, we also show that ena is sufficient to provide FCs with the

ability to produce filopodia even outside the PFC domain (Fig.8E). We propose that filopodia could act in two different non-exclusive ways to participate in oskar mRNA transport towards the oocyte posterior cortex. These hypotheses were already discussed in the discussion section entitled “putative roles of filopodia and E-Cadherin at the PAC-oocyte interface”. We propose that the first filopodia role is to maintain oocyte and PAC membranes tight for longer than in the lateral region. It is based on our results showing that gain of *ena* is sufficient to keep oocyte and follicle cell membranes tight in a larger posterior region of follicles upon *ena* ectopic expression, as shown in Fig.8A,A'. We also examined earlier follicles when *ena* is expressed in all PFCs, and we found that at stage 8, all PFCs in tight contact with the oocyte express *ena* and produce filopodia penetrating the oocyte. We added this result in Supplementary Fig.6E. This, together with our previous result showing *ena* expression gets restricted to PACs, the only FCs in tight contact with the oocyte at stage 10, confirm the correlation between *ena* expression and the maintenance of a tight PFCoocyte membrane interface. The second role that we discussed was only speculative and based on the literature. We proposed that filopodia could provide more cortical membrane surface in the oocyte that would concentrate actin proteins interacting with oskar mRNA particles to anchor and trap them and/or protect them from cytoplasmic flows that could detach them. To test this hypothesis further, we performed electron microscopy on follicles expressing *oskar::GFP* to visualize oskar mRNA particles containing Oskar protein with the APEX-GBP system. We found cases in which oskar mRNA particles were detected in the adjacent section where filopodia can be observed. Since sections are 70 nm thick, it indicates that oskar mRNA particles can be found in close contact with filopodia. In other cases, we detected oskar mRNA particles lying at the cortex between filopodia, suggesting that filopodia could also exert a protection of oskar mRNA anchoring by trapping particles in a rough surface. These new results have been added in Fig.6D-E'.

REVIEWERS' COMMENTS

Reviewer #1 (Remarks to the Author):

I appreciate the effort made by the authors to answer all my questions. The new results following the reviewers' comments significantly improve their article. Consequently, in my opinion, the article can be accepted for publication as is.

Reviewer #2 (Remarks to the Author):

The additional data has improved this manuscript and I think that it is appropriate for publication.

Reviewer #3 (Remarks to the Author):

With these revisions to the manuscript and the thorough response to the reviewer comments, the authors have fully addressed my concerns. I support publication in of this manuscript in its current form.